# Certified Robustness for Top-$k$ Predictions against Adversarial Perturbations via Randomized Smoothing

**Jinyuan Jia, Xiaoyu Cao, Binghui Wang, Neil Zhenqiang Gong**
Duke University
{jinyuan.jia,xiaoyu.cao,binghui.wang,neil.gong}@duke.edu

## Abstract

It is well-known that classifiers are vulnerable to adversarial perturbations. To defend against adversarial perturbations, various *certified* robustness results have been derived. However, existing certified robustnesses are limited to top-1 predictions. In many real-world applications, top-$k$ predictions are more relevant. In this work, we aim to derive certified robustness for top-$k$ predictions. In particular, our certified robustness is based on *randomized smoothing*, which turns any classifier to a new classifier via adding noise to an input example. We adopt randomized smoothing because it is scalable to large-scale neural networks and applicable to any classifier. We derive a *tight* robustness in $\ell_2$ norm for top-$k$ predictions when using randomized smoothing with Gaussian noise. We find that generalizing the certified robustness from top-1 to top-$k$ predictions faces significant technical challenges. We also empirically evaluate our method on CI-FAR10 and ImageNet. For example, our method can obtain an ImageNet classifier with a certified top-5 accuracy of 62.8% when the $\ell_2$-norms of the adversarial perturbations are less than 0.5 (=127/255). Our code is publicly available at: https://github.com/jjy1994/Certify_Topk.

## 1 Introduction

Classifiers are vulnerable to adversarial perturbations (Szegedy et al., 2014; Goodfellow et al., 2015; Carlini & Wagner, 2017b; Jia & Gong, 2018). Specifically, given an example $\mathbf{x}$ and a classifier $f$, an attacker can carefully craft a perturbation $\delta$ such that $f$ makes predictions for $\mathbf{x} + \delta$ as the attacker desires. Various empirical defenses (e.g., Goodfellow et al. (2015); Svoboda et al. (2019); Buckman et al. (2018); Ma et al. (2018); Guo et al. (2018); Dhillon et al. (2018); Xie et al. (2018); Song et al. (2018)) have been proposed to defend against adversarial perturbations. However, these empirical defenses were often soon broken by adaptive adversaries (Carlini & Wagner, 2017a; Athalye et al., 2018). As a response, *certified* robustness (e.g., Wong & Kolter (2018); Raghunathan et al. (2018a); Liu et al. (2018); Lecuyer et al. (2019); Cohen et al. (2019)) against adversarial perturbations has been developed. In particular, a robust classifier verifiably predicts the same top-1 label for data points in a certain region around any example $\mathbf{x}$.

In many applications such as recommender systems, web search, and image classification cloud service (Clarifai; Google Cloud Vision), top-$k$ predictions are more relevant. In particular, given an example, a set of $k$ most likely labels are predicted for the example. However, existing certified robustness results are limited to top-1 predictions, leaving top-$k$ robustness unexplored. To bridge this gap, we study certified robustness for top-$k$ predictions in this work. Our certified top-$k$ robustness leverages *randomized smoothing* (Cao & Gong, 2017; Cohen et al., 2019), which turns any base classifier $f$ to be a robust classifier via adding random noise to an example. For instance, Cao & Gong (2017) is the first to propose randomized smoothing with uniform noise as an empirical defense. We consider random Gaussian noise because of its certified robustness guarantee (Cohen et al., 2019). Specifically, we denote by $p_i$ the probability that the base classifier $f$ predicts label $i$ for the Gaussian random variable $\mathcal{N}(\mathbf{x}, \sigma^2 I)$. The smoothed classifier $g_k(\mathbf{x})$ predicts the $k$ labels with the largest probabilities $p_i$'s for the example $\mathbf{x}$. We adopt randomized smoothing because it is scalable to large-scale neural networks and applicable to any base classifier.

Our major theoretical result is a tight certified robustness bound for top-$k$ predictions when using randomized smoothing with Gaussian noise. Specifically, given an example $\mathbf{x}$, a label $l$ is verifiably among the top-$k$ labels predicted by the smoothed classifier $g_k(\mathbf{x} + \delta)$ when the $\ell_2$-norm of the adversarial perturbation $\delta$ is less than a threshold (called *certified radius*). The certified radius for top-1 predictions derived by Cohen et al. (2019) is a special case of our certified radius when $k = 1$. As our results and proofs show, generalizing certified robustness from top-1 to top-$k$ predictions faces significant new challenges and requires new techniques. Our certified radius is the unique solution to an equation, which depends on $\sigma$, $p_l$, and the $k$ largest probabilities $p_i$'s (excluding $p_l$). However, computing our certified radius in practice faces two challenges: 1) it is hard to exactly compute the probability $p_l$ and the $k$ largest probabilities $p_i$'s, and 2) the equation about the certified radius does not have an analytical solution. To address the first challenge, we estimate *simultaneous confidence intervals* of the label probabilities via the Clopper-Pearson method and *Bonferroni correction* in statistics. To address the second challenge, we propose an algorithm to solve the equation to obtain a lower bound of the certified radius, where the lower bound can be tuned to be arbitrarily close to the true certified radius. We evaluate our method on CIFAR10 (Krizhevsky & Hinton, 2009) and ImageNet (Deng et al., 2009) datasets. For instance, on ImageNet, our method respectively achieves approximate certified top-1, top-3, and top-5 accuracies as 46.6%, 57.8%, and 62.8% when the $\ell_2$-norms of the adversarial perturbations are less than 0.5 (127/255) and $\sigma = 0.5$.

Our contributions are summarized as follows:

- **Theory.** We derive the first certified radius for top-$k$ predictions. Moreover, we prove our certified radius is tight for randomized smoothing with Gaussian noise.
- **Algorithm.** We develop algorithms to estimate our certified radius in practice.
- **Evaluation.** We empirically evaluate our method on CIFAR10 and ImageNet.

## 2 CERTIFIED RADIUS FOR TOP-$k$ PREDICTIONS

Suppose we have a base classifier $f$, which maps an example $\mathbf{x} \in \mathbb{R}^d$ to one of $c$ candidate labels $\{1, 2, \cdots, c\}$. $f$ can be any classifier. Randomized smoothing (Cohen et al., 2019) adds an isotropic Gaussian noise $\mathcal{N}(0, \sigma^2 I)$ to an example $\mathbf{x}$. We denote $p_i$ as the probability that the base classifier $f$ predicts label $i$ when adding a random isotropic Gaussian noise $\epsilon$ to the example $\mathbf{x}$, i.e., $p_i = \Pr(f(\mathbf{x} + \epsilon) = i)$, where $\epsilon \sim \mathcal{N}(0, \sigma^2 I)$. The smoothed classifier $g_k(\mathbf{x})$ returns the set of $k$ labels with the largest probabilities $p_i$'s when taking an example $\mathbf{x}$ as input. Our goal is to derive a certified radius $R_l$ such that we have $l \in g_k(\mathbf{x} + \delta)$ for all $||\delta||_2 < R_l$. Our main theoretical results are summarized in the following two theorems.

**Theorem 1** (Certified Radius for Top-$k$ Predictions). *Suppose we are given an example $\mathbf{x}$, an arbitrary base classifier $f$, $\epsilon \sim \mathcal{N}(0, \sigma^2 I)$, a smoothed classifier $g$, an arbitrary label $l \in \{1, 2, \cdots, c\}$, and $\underline{p}_l, \overline{p}_1, \cdots, \overline{p}_{l-1}, \overline{p}_{l+1}, \cdots, \overline{p}_c \in [0, 1]$ that satisfy the following conditions:*

$$Pr(f(\mathbf{x} + \epsilon) = l) \geq \underline{p}_l \text{ and } Pr(f(\mathbf{x} + \epsilon) = i) \leq \overline{p}_i, \forall i \neq l, \tag{1}$$

*where $\underline{p}$ and $\overline{p}$ indicate lower and upper bounds of $p$, respectively. Let $\overline{p}_{b_k} \geq \overline{p}_{b_{k-1}} \geq \cdots \geq \overline{p}_{b_1}$ be the $k$ largest ones among $\{\overline{p}_1, \cdots, \overline{p}_{l-1}, \overline{p}_{l+1}, \cdots, \overline{p}_c\}$, where ties are broken uniformly at random. Moreover, we denote by $S_t = \{b_1, b_2, \cdots, b_t\}$ the set of $t$ labels with the smallest probability upper bounds in the $k$ largest ones and by $\overline{p}_{S_t} = \sum_{j=1}^t \overline{p}_{b_j}$ the sum of the $t$ probability upper bounds, where $t = 1, 2, \cdots, k$. Then, we have:*

$$l \in g_k(\mathbf{x} + \delta), \forall ||\delta||_2 < R_l, \tag{2}$$

*where $R_l$ is the unique solution to the following equation:*

$$\Phi(\Phi^{-1}(\underline{p}_l) - \frac{R_l}{\sigma})) - \min_{t=1}^{k} \frac{\Phi(\Phi^{-1}(\overline{p}_{S_t}) + \frac{R_l}{\sigma}))}{t} = 0, \tag{3}$$

*where $\Phi$ and $\Phi^{-1}$ are the cumulative distribution function and its inverse of the standard Gaussian distribution, respectively.*

*Proof.* See Appendix A. □

---

**Algorithm 1:** PREDICT

---

**Input:** $f$, $k$, $\sigma$, $\mathbf{x}$, $n$, and $\alpha$.
**Output:** ABSTAIN or predicted top-$k$ labels.

1  $T = \emptyset$
2  counts = SAMPLEUNDERNOISE$(f, \sigma, \mathbf{x}, n)$
3  $c_1, c_2, \cdots, c_{k+1} = $ top-$\{k+1\}$ indices in counts (ties are broken uniformly at random)
4  $n_{c_1}, n_{c_2}, \cdots, n_{c_{k+1}} = $ counts$[c_1]$, counts$[c_2]$, $\cdots$, counts$[c_{k+1}]$
5  **for** $t \leftarrow 1$ **to** $k$ **do**
6     **if** BINOMPVALUE$(n_{c_t}, n_{c_t} + n_{c_{t+1}}, 0.5) \leq \alpha$ **then**
7        | $\quad T = T \cup c_t$
8     **else**
9        | $\quad$ **return** ABSTAIN
10 **return** $T$

---

**Theorem 2** (Tightness of the Certified Radius). *Assuming we have $\underline{p_l} + \sum_{j=1}^{k} \overline{p}_{b_j} \leq 1$ and $\underline{p_l} + \sum_{i=1,\cdots,l-1,l+1,\cdots,c} \overline{p}_i \geq 1$. Then, for any perturbation $||\delta||_2 > R_l$, there exists a base classifier $f^*$ consistent with (1) but we have $l \notin g_k(\mathbf{x} + \delta)$.*

*Proof.* We show a proof sketch here. Our detailed proof is in Appendix B. In our proof, we first show that, via *mathematical induction* and the *intermediate value theorem*, we can construct $k + 1$ disjoint regions $\mathcal{C}_i, i \in \{l\} \cup \{b_1, b_2, \cdots, b_k\}$ that satisfy $\Pr(\mathbf{x} + \epsilon \in \mathcal{C}_l) = \underline{p_l}$, $\Pr(\mathbf{x} + \epsilon \in \mathcal{C}_i) = \overline{p}_i$, and $\Pr(\mathbf{x} + \delta + \epsilon \in \mathcal{C}_i)$ is no smaller than some critical value for $i \in \{b_1, b_2, \cdots, b_k\}$. Moreover, we divide the remaining region $\mathbb{R}^d \setminus (\cup_{i=l,b_1,b_2,\cdots,b_k} \mathcal{C}_i)$ into $c - k - 1$ regions, which we denote as $\mathcal{C}_{b_{k+1}}, \mathcal{C}_{b_{k+2}}, \cdots, \mathcal{C}_{b_{c-1}}$ and satisfy $\Pr(\mathbf{x} + \epsilon \in \mathcal{C}_i) \leq \overline{p}_i$ for $i = b_{k+1}, b_{k+2}, \cdots, b_{c-1}$. Then, we construct a base classifier $f^*$ that predicts label $i$ for an example if and only if the example is in the region $\mathcal{C}_i$, where $i \in \{1, 2, \cdots, c\}$. As $\underline{p_l} + \sum_{j=1}^{k} \overline{p}_{b_j} \leq 1$ and $\underline{p_l} + \sum_{i=1,\cdots,l-1,l+1,\cdots,c} \overline{p}_i \geq 1$, $f^*$ is well-defined. Moreover, $f^*$ satisfies the conditions in (1). Finally, we show that if $||\delta||_2 > R_l$, then we have $\Pr(f^*(\mathbf{x} + \delta + \epsilon) = l) < \min_{j=1}^{k} \Pr(f^*(\mathbf{x} + \delta + \epsilon) = b_j)$, i.e., $l \notin g_k(\mathbf{x} + \delta)$. $\qquad\square$

We have several observations about our theorems.

- Our certified radius is applicable to any base classifier $f$.

- According to Equation 3, our certified radius $R_l$ depends on $\sigma$, $\underline{p_l}$, and the $k$ largest probability upper bounds $\{\overline{p}_{b_k}, \overline{p}_{b_{k-1}}, \cdots, \overline{p}_{b_1}\}$ excluding $\overline{p}_l$. When the lower bound $\underline{p_l}$ and the upper bounds $\{\overline{p}_{b_k}, \overline{p}_{b_{k-1}}, \cdots, \overline{p}_{b_1}\}$ are tighter, the certified radius $R_l$ is larger. When $R_l < 0$, the label $l$ is not among the top-$k$ labels predicted by the smoothed classifier even if no perturbation is added, i.e., $l \notin g_k(\mathbf{x})$.

- When using randomized smoothing with Gaussian noise and no further assumptions are made on the base classifier, it is impossible to certify a $\ell_2$ radius for top-$k$ predictions that is larger than $R_l$.

- When $k = 1$, we have $R_l = \frac{\sigma}{2}(\Phi^{-1}(\underline{p_l}) - \Phi^{-1}(\overline{p}_{b_1}))$, where $\overline{p}_{b_1}$ is an upper bound of the largest label probability excluding $\overline{p}_l$. The certified radius derived by Cohen et al. (2019) for top-1 predictions (i.e., their Equation 3) is a special case of our certified radius with $k = 1$, $l = A$, and $b_1 = B$.

## 3 PREDICTION AND CERTIFICATION IN PRACTICE

### 3.1 PREDICTION

It is challenging to compute the top-$k$ labels $g_k(\mathbf{x})$ predicted by the smoothed classifier, because it is challenging to compute the probabilities $p_i$'s exactly. To address the challenge, we resort to a Monte Carlo method that predicts the top-$k$ labels with a probabilistic guarantee. In particular, we leverage the hypothesis testing result from a recent work (Hung et al., 2019). Algorithm 1

shows our PREDICT function to estimate the top-$k$ labels predicted by the smoothed classifier. The function SAMPLEUNDERNOISE$(f, \sigma, \mathbf{x}, n)$ first randomly samples $n$ noise $\epsilon_1, \epsilon_2, \cdots, \epsilon_n$ from the Gaussian distribution $\mathcal{N}(0, \sigma^2 I)$, uses the base classifier $f$ to predict the label of $\mathbf{x} + \epsilon_j$ for each $j \in \{1, 2, \cdots, n\}$, and returns the frequency of each label, i.e., counts$[i] = \sum_{j=1}^{n} \mathbb{I}(f(\mathbf{x} + \epsilon_j) = i)$ for $i \in \{1, 2, \cdots, c\}$. The function BINOMPVALUE performs the hypothesis testing to calibrate the abstention threshold such that we can bound with probability $\alpha$ of returning an incorrect set of top-$k$ labels. Formally, we have the following proposition:

**Proposition 1.** *With probability at least $1 - \alpha$ over the randomness in* PREDICT, *if* PREDICT *returns a set $T$ (i.e., does not ABSTAIN), then we have $g_k(\mathbf{x}) = T$.*

*Proof.* See Appendix C. □

## 3.2 CERTIFICATION

Given a base classifier $f$, an example $\mathbf{x}$, a label $l$, and the standard deviation $\sigma$ of the Gaussian noise, we aim to compute the certified radius $R_l$. According to our Equation 3, our $R_l$ relies on a lower bound of $p_l$, i.e., $\underline{p_l}$, and the upper bound of $p_{S_t}$, i.e., $\overline{p}_{S_t}$, which are related to $f$, $\mathbf{x}$, and $\sigma$. We first discuss two Monte Carlo methods to estimate $\underline{p_l}$ and $\overline{p}_{S_t}$ with probabilistic guarantees. However, given $\underline{p_l}$ and $\overline{p}_{S_t}$, it is still challenging to exactly solve $R_l$ as the Equation 3 does not have an analytical solution. To address the challenge, we design an algorithm to obtain a lower bound of $R_l$ via solving Equation 3 through binary search. Our lower bound can be tuned to be arbitrarily close to $R_l$.

### 3.2.1 ESTIMATING $\underline{p_l}$ AND $\overline{p}_{S_t}$

Our approach has two steps. The first step is to estimate $\underline{p_l}$ and $\overline{p}_i$ for $i \neq l$. The second step is to estimate $\overline{p}_{S_t}$ using $\overline{p}_i$ for $i \neq l$.

**Estimating $\underline{p_l}$ and $\overline{p}_i$ for $i \neq l$:** The probabilities $p_1, p_2, \cdots, p_c$ can be viewed as a multinomial distribution over the labels $\{1, 2, \cdots, c\}$. If we sample a Gaussian noise $\epsilon$ uniformly at random, then the label $f(\mathbf{x} + \epsilon)$ can be viewed as a sample from the multinomial distribution. Therefore, estimating $\underline{p_l}$ and $\overline{p}_i$ for $i \neq l$ is essentially a one-sided *simultaneous confidence interval* estimation problem. In particular, we aim to estimate these bounds with a confidence level at least $1 - \alpha$. In statistics, Goodman (1965); Sison & Glaz (1995) are well-known methods for simultaneous confidence interval estimations. However, these methods are insufficient for our problem. Specifically, Goodman's method is based on Chi-square test, which requires the expected count for each label to be no less than 5. We found that this is usually not satisfied, e.g., ImageNet has 1,000 labels, some of which have close-to-zero probabilities and do not have more than 5 counts even if we sample a large number of Gaussian noise. Sison & Glaz's method guarantees a confidence level of *approximately* $1 - \alpha$, which means that the confidence level could be (slightly) smaller than $1 - \alpha$. However, we aim to achieve a confidence level of at least $1 - \alpha$. To address these challenges, we discuss two confidence interval estimation methods as follows:

**1) BinoCP.** This method estimates $\underline{p_l}$ using the standard one-sided Clopper-Pearson method and treats $\overline{p}_i$ as $\overline{p}_i = 1 - \underline{p_l}$ for each $i \neq l$. Specifically, we sample $n$ random noise from $\mathcal{N}(0, \sigma I^2)$, i.e., $\epsilon_1, \epsilon_2, \cdots, \epsilon_n$. We denote the count for the label $l$ as $n_l = \sum_{j=1}^{n} \mathbb{I}(f(\mathbf{x} + \epsilon_j) = l)$. $n_l$ follows a binomial distribution with parameters $n$ and $p_l$, i.e., $n_l \sim Bin(n, p_l)$. Therefore, according to the Clopper-Pearson method, we have:

$$\underline{p_l} = B(\alpha; n_l, n - n_l + 1), \tag{4}$$

where $1 - \alpha$ is the confidence level and $B(\alpha; u, v)$ is the $\alpha$th quantile of the Beta distribution with shape parameters $u$ and $v$. We note that the Clopper-Pearson method was also adopted by Cohen et al. (2019) to estimate label probability for their certified radius of top-1 predictions.

**2) SimuEM.** The above method estimates $\overline{p}_i$ as $1 - \underline{p_l}$, which may be conservative. A conservative estimation makes the certified radius smaller than what it should be. Therefore, we introduce SimuEM to directly estimate $\overline{p}_i$ together with $\underline{p_l}$. We let $n_i = \sum_{j=1}^{n} \mathbb{I}(f(\mathbf{x} + \epsilon_j) = i)$ for each $i \in \{1, 2, \cdots, c\}$. Each $n_i$ follows a binomial distribution with parameters $n$ and $p_i$. We first use

---

**Algorithm 2:** CERTIFY

---

**Input:** $f$, $k$, $\sigma$, $\mathbf{x}$, $l$, $n$, $\mu$, and $\alpha$.
**Output:** ABSTAIN or $\underline{R_l}$.

1   counts = SAMPLEUNDERNOISE($f, \sigma, \mathbf{x}, n, \alpha$)
2   $[\underline{p_l}, \overline{p}_1, \cdots, \overline{p}_{l-1}, \overline{p}_{l+1}, \cdots, \overline{p}_c]$ = BINOCP(counts, $\alpha$) or SIMUEM(counts, $\alpha$)
3   $\underline{R_l} = 0$
4   **for** $t \leftarrow 1$ **to** $k$ **do**
5      $\overline{p}_{S_t} = \min(\sum_{j=1}^{t} \overline{p}_{b_j}, 1 - \underline{p_l})$
6      $\underline{R_l}^t =$ BINARYSEARCH( $\underline{p_l}, \overline{p}_{S_t}, t, \sigma, \mu$ )
7      **if** $\underline{R_l}^t > \underline{R_l}$ **then**
8         $\underline{R_l} = \underline{R_l}^t$
9   **if** $\underline{R_l} > 0$ **then**
10      **return** $\underline{R_l}$
11   **else**
12      **return** ABSTAIN

---

the Clopper-Pearson method to estimate a one-sided confidence interval for each label $i$, and then we obtain simultaneous confidence intervals by leveraging the *Bonferroni correction*. Specifically, if we can obtain a confidence interval with confidence level at least $1 - \frac{\alpha}{c}$ for each label $i$, then *Bonferroni correction* tells us that the overall confidence level for the simultaneous confidence intervals is at least $1 - \alpha$, i.e., we have confidence level at least $1 - \alpha$ that all confidence intervals hold at the same time. Formally, we have the following bounds by applying the Clopper-Pearson method with confidence level $1 - \frac{\alpha}{c}$ to each label:

$$\underline{p_l} = B\left(\frac{\alpha}{c}; n_l, n - n_l + 1\right) \tag{5}$$

$$\overline{p}_i = B(1 - \frac{\alpha}{c}; n_i + 1, n - n_i), \ \forall i \neq l. \tag{6}$$

**Estimating $\overline{p}_{S_t}$:** One natural method is to estimate $\overline{p}_{S_t} = \sum_{j=1}^{t} \overline{p}_{b_j}$. However, this bound may be loose. For example, when using **BinoCP** to estimate the probability bounds, we have $\overline{p}_{S_t} = t \cdot (1 - \underline{p_l})$, which may be bigger than 1. To address the challenge, we derive another bound for $\overline{p}_{S_t}$ from another perspective. Specifically, we have $p_{S_t} \leq \sum_{i \neq l} p_i \leq 1 - \underline{p_l}$. Therefore, we can use $1 - \underline{p_l}$ as an upper bound of $p_{S_t}$, i.e., $\overline{p}_{S_t} = 1 - \underline{p_l}$. Finally, we combine the above two estimations by taking the minimal one, i.e., $\overline{p}_{S_t} = \min(\sum_{j=1}^{t} \overline{p}_{b_j}, 1 - \underline{p_l})$.

### 3.2.2 ESTIMATING A LOWER BOUND OF THE CERTIFIED RADIUS $R_l$

It is challenging to compute the certified radius $R_l$ exactly because Equation 3 does not have an analytical solution. To address the challenge, we design a method to estimate a lower bound of $R_l$ that can be tuned to be arbitrarily close to $R_l$. Specifically, we first approximately solve the following equation for each $t \in \{1, 2, \cdots, k\}$:

$$\Phi(\Phi^{-1}(\underline{p_l}) - \frac{R_l^t}{\sigma})) - \frac{\Phi(\Phi^{-1}(\overline{p}_{S_t}) + \frac{R_l^t}{\sigma}))}{t} = 0. \tag{7}$$

We note that it is still difficult to obtain an analytical solution to Equation 7 when $t > 1$. However, we notice that the left-hand side has the following properties: 1) it decreases as $R_l^t$ increases; 2) when $R_l^t \rightarrow -\infty$, it is greater than 0; 3) when $R_l^t \rightarrow \infty$, it is smaller than 0. Therefore, there exists a unique solution $R_l^t$ to Equation 7. Moreover, we leverage binary search to find a lower bound $\underline{R_l}^t$ that can be arbitrarily close to the exact solution $R_l^t$. In particular, we run the binary search until the left-hand side of Equation 7 is non-negative and the width of the search interval is less than a parameter $\mu > 0$. Formally, we have:

$$\underline{R_l}^t \leq R_l^t \leq \underline{R_l}^t + \mu, \ \forall t \in \{1, 2, \cdots, k\}. \tag{8}$$

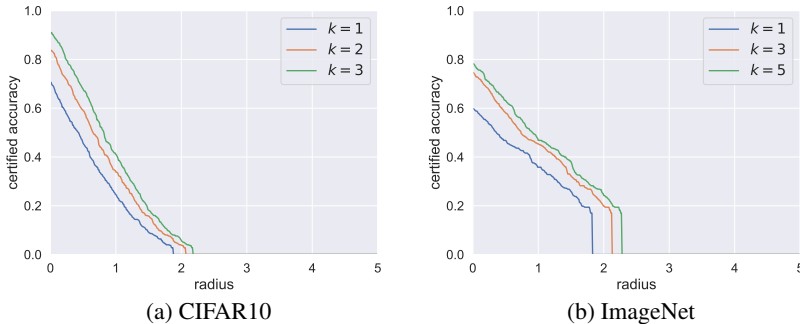

(a) CIFAR10            (b) ImageNet

**Figure 1: Impact of $k$ on the certified top-$k$ accuracy.**

After obtaining $\underline{R_l}^t$, we let $\underline{R_l} = \max_{t=1}^k \underline{R_l}^t$ be our lower bound of $R_l$. Based on $R_l = \max_{t=1}^k R_l^t$ and Equation 8, we have the following guarantee:

$$\underline{R_l} \leq R_l \leq \underline{R_l} + \mu. \tag{9}$$

### 3.2.3 COMPLETE CERTIFICATION ALGORITHM

Algorithm 2 shows our algorithm to estimate the certified radius for a given example $\mathbf{x}$ and a label $l$. The function SAMPLEUNDERNOISE is the same as in Algorithm 1. Functions BINOCP and SIMUEM return the estimated probability bound for each label. Function BINARYSEARCH performs binary search to solve the Equation 7 and returns a solution satisfying Equation 8. Formally, our algorithm has the following guarantee:

**Proposition 2.** *With probability at least $1-\alpha$ over the randomness in* CERTIFY*, if* CERTIFY *returns a radius $\underline{R_l}$ (i.e., does not ABSTAIN), then we have $l \in g_k(\mathbf{x} + \delta), \forall ||\delta||_2 < \underline{R_l}$.*

*Proof.* See Appendix D. □

## 4 EXPERIMENTS

### 4.1 EXPERIMENTAL SETUP

**Datasets and models:** We conduct experiments on the standard CIFAR10 (Krizhevsky & Hinton, 2009) and ImageNet (Deng et al., 2009) datasets to evaluate our method. We use the publicly available pre-trained models from Cohen et al. (2019). Specifically, the architectures of the base classifiers are ResNet-110 and ResNet-50 for CIFAR10 and ImageNet, respectively.

**Parameter setting:** We study the impact of $k$, the confidence level $1 - \alpha$, the noise level $\sigma$, the number of samples $n$, and the confidence interval estimation methods on the certified radius. Unless otherwise mentioned, we use the following default parameters: $k = 3$, $\alpha = 0.001$, $\sigma = 0.5$, $n = 100,000$, and $\mu = 10^{-5}$. Moreover, we use **SimuEM** to estimate bounds of label probabilities. When studying the impact of one parameter on the certified radius, we fix the other parameters to their default values.

**Approximate certified top-$k$ accuracy:** For each testing example $\mathbf{x}$ whose true label is $l$, we compute the certified radius $\underline{R_l}$ using the CERTIFY algorithm. Then, we compute the certified top-$k$ accuracy at a radius $r$ as the fraction of testing examples whose certified radius are at least $r$. Note that our computed certified top-$k$ accuracy is an *approximate certified top-$k$ accuracy* instead of the true certified top-$k$ accuracy. However, we can obtain a lower bound of the true certified top-$k$ accuracy based on the approximate certified top-$k$ accuracy. Appendix E shows the details. Moreover, the gap between the lower bound of the true certified top-$k$ accuracy and the approximate top-$k$ accuracy is negligible when $\alpha$ is small. For convenience, we simply use the term certified top-$k$ accuracy in the paper.

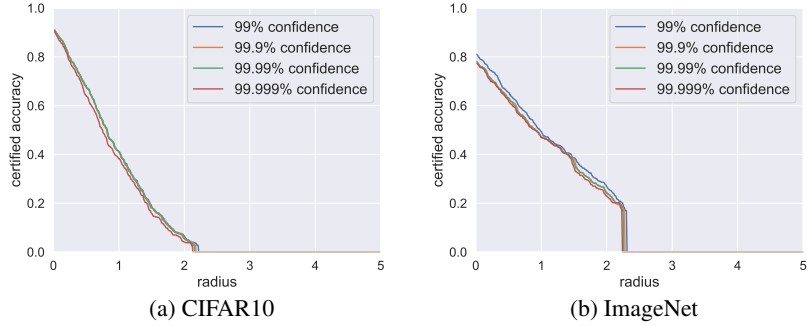

**Figure 2: Impact of the confidence level $1 - \alpha$ on the certified top-$3$ accuracy.**

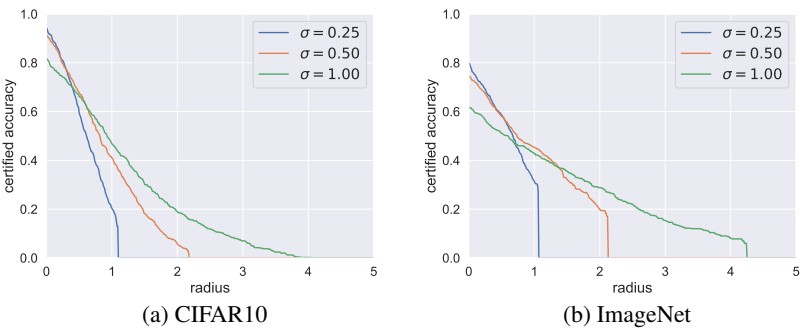

**Figure 3: Impact of $\sigma$ on the certified top-$3$ accuracy.**

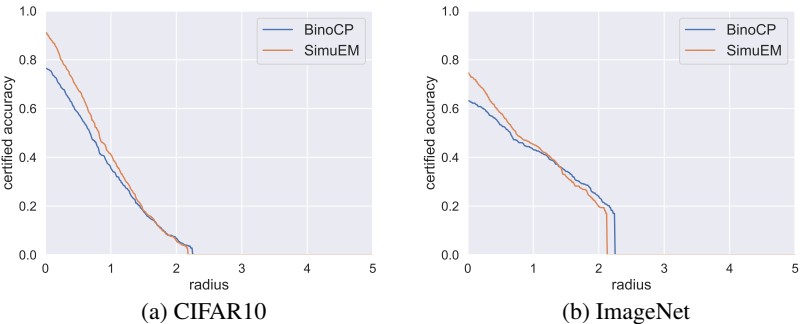

**Figure 4: BinoCP vs. SimuEM, where $k = 3$.**

## 4.2 EXPERIMENTAL RESULTS

Figure 1 shows the certified top-$k$ accuracy as the radius $r$ increases for different $k$. Naturally, the certified top-$k$ accuracy increases as $k$ increases. On CIFAR10, we respectively achieve certified top-1, top-2, and top-3 accuracies as 45.2%, 58.8%, and 67.2% when the $\ell_2$-norm of the adversarial perturbation is less than 0.5 (127/255). On ImageNet, we respectively achieve certified top-1, top-3, and top-5 accuracies as 46.6%, 57.8%, and 62.8% when the $\ell_2$-norm of the adversarial perturbation is less than 0.5. On CIFAR10, the gaps between the certified top-$k$ accuracy for different $k$ are smaller than those between the top-$k$ accuracy under no attacks, and they become smaller as the radius increases. On ImageNet, the gaps between the certified top-$k$ accuracy for different $k$ remain similar to those between the top-$k$ accuracy under no attacks as the radius increases. Figure 2 shows the influence of the confidence level. We observe that confidence level has a small influence on the certified top-$k$ accuracy as the different curves almost overlap. The reason is that the estimated confidence intervals of the probabilities shrink slowly as the confidence level increases. Figure 3 shows the influence of $\sigma$. We observe that $\sigma$ controls a trade-off between normal accuracy under no attacks and robustness. Specifically, when $\sigma$ is smaller, the accuracy under no attacks (i.e., the accuracy when radius is 0) is larger, but the certified top-$k$ accuracy drops more quickly as the radius

increases. Figure 4 compares **BinoCP** with **SimuEM**. The results show that **SimuEM** is better when the certified radius is small, while **BinoCP** is better when the certified radius is large. We found the reason is that when the certified radius is large, $\underline{p_l}$ is relatively large, and thus $1 - \underline{p_l}$ already provides a good estimation for $\overline{p}_i$, where $i \neq l$.

## 5 RELATED WORK

Numerous defenses have been proposed against adversarial perturbations in the past several years. These defenses either show robustness against existing attacks empirically, or prove the robustness against arbitrary bounded-perturbations (known as *certified defenses*).

### 5.1 EMPIRICAL DEFENSES

The community has proposed many empirical defenses. The most effective empirical defense is *adversarial training* (Goodfellow et al., 2015; Kurakin et al., 2017; Tramèr et al., 2018; Madry et al., 2018). However, adversarial training does not have certified robustness guarantees. Other examples of empirical defenses include defensive distillation (Papernot et al., 2016), MagNet (Meng & Chen, 2017), PixelDefend (Song et al., 2017), Feature squeezing (Xu et al., 2018), and many others (Liu et al., 2019; Svoboda et al., 2019; Schott et al., 2019; Buckman et al., 2018; Ma et al., 2018; Guo et al., 2018; Dhillon et al., 2018; Xie et al., 2018; Song et al., 2018; Samangouei et al., 2018; Na et al., 2018; Metzen et al., 2017). However, many of these defenses were soon broken by adaptive attacks (Carlini & Wagner, 2017a; Athalye et al., 2018; Uesato et al., 2018; Athalye & Carlini, 2018).

### 5.2 CERTIFIED DEFENSES

To end the arms race between defenders and adversaries, researchers have developed certified defenses against adversarial perturbations. Specifically, in a certifiably robust classifier, the predicted top-1 label is verifiably constant within a certain region (e.g., $\ell_2$-norm ball) around an input example, which provides a lower bound of the adversarial perturbation. Such certified defenses include satisfiability modulo theories based methods (Katz et al., 2017; Carlini et al., 2017; Ehlers, 2017; Huang et al., 2017), mixed integer linear programming based methods (Cheng et al., 2017; Lomuscio & Maganti, 2017; Dutta et al., 2017; Fischetti & Jo, 2018; Bunel et al., 2018), abstract interpretation based methods (Gehr et al., 2018; Tjeng et al., 2018), and global (or local) Lipschitz constant based methods (Cisse et al., 2017; Gouk et al., 2018; Tsuzuku et al., 2018; Anil et al., 2019; Wong & Kolter, 2018; Wang et al., 2018a;b; Raghunathan et al., 2018a;b; Wong et al., 2018; Dvijotham et al., 2018a;b; Croce et al., 2018; Gehr et al., 2018; Mirman et al., 2018; Singh et al., 2018; Gowal et al., 2018; Weng et al., 2018; Zhang et al., 2018). However, these methods are not scalable to large neural networks and/or make assumptions on the architectures of the neural networks. For example, these defenses are not scalable/applicable to the complex neural networks for ImageNet.

Randomized smoothing was first proposed as an empirical defense (Cao & Gong, 2017; Liu et al., 2018) without deriving the certified robustness guarantees. For instance, Cao & Gong (2017) proposed randomized smoothing with uniform noise from a hypercube centered at an example. Lecuyer et al. (2019) was the first to prove the certified robustness guarantee of randomized smoothing for top-1 predictions. Their results leverage differential privacy. Subsequently, Li et al. (2018) further leverages information theory to improve the certified radius bound. Cohen et al. (2019) obtains a tight certified radius bound for randomized smoothing with Gaussian noise by leveraging the Neyman-Pearson Lemma. Pinot et al. (2019) theoretically demonstrated the robustness to adversarial attacks of randomized smoothing when adding noise from Exponential family distributions and devised an upper bound on the adversarial generalization gap of randomized neural networks. Lee et al. (2019) generalized randomized smoothing to discrete data. Salman et al. (2019) employed adversarial training to improve the performance of randomized smoothing. Unlike the other certified defenses, randomized smoothing is scalable to large neural networks and applicable to arbitrary classifiers. Our work derives the first certified robustness guarantee of randomized smoothing for top-$k$ predictions. Moreover, we show that our robustness guarantee is tight for randomized smoothing with Gaussian noise.

## 6    CONCLUSION

Adversarial perturbation poses a fundamental security threat to classifiers. Existing certified defenses focus on top-1 predictions, leaving top-$k$ predictions untouched. In this work, we derive the first certified radius under $\ell_2$-norm for top-$k$ predictions. Our results are based on randomized smoothing. Moreover, we prove that our certified radius is tight for randomized smoothing with Gaussian noise. In order to compute the certified radius in practice, we further propose simultaneous confidence interval estimation methods as well as design an algorithm to estimate a lower bound of the certified radius. Interesting directions for future work include 1) deriving a tight certified radius under other norms such as $\ell_1$ and $\ell_\infty$, 2) studying which noise gives the tightest certified radius for randomized smoothing, and 3) studying certified robustness for top-$k$ ranking.

**ACKNOWLEDGMENTS**
We thank the anonymous reviewers for insightful reviews. This work was supported by NSF grant No. 1937786.

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

## A    PROOF OF THEOREM 1

Given an example $\mathbf{x}$, we define the following two random variables:

$$\mathbf{X} = \mathbf{x} + \epsilon \sim \mathcal{N}(\mathbf{x}, \sigma^2 I), \tag{10}$$

$$\mathbf{Y} = \mathbf{x} + \delta + \epsilon \sim \mathcal{N}(\mathbf{x} + \delta, \sigma^2 I), \tag{11}$$

where $\epsilon \sim \mathcal{N}(0, \sigma^2 I)$. The random variables $\mathbf{X}$ and $\mathbf{Y}$ represent random samples obtained by adding isotropic Gaussian noise to the example $\mathbf{x}$ and its perturbed version $\mathbf{x} + \delta$, respectively. Cohen et al. (2019) applied the standard Neyman-Pearson Lemma (Neyman & Pearson, 1933) to the above two random variables, and obtained the following lemma:

**Lemma 1** (Neyman-Pearson for Gaussians with different means). *Let $\mathbf{X} \sim \mathcal{N}(\mathbf{x}, \sigma^2 I)$, $\mathbf{Y} \sim \mathcal{N}(\mathbf{x} + \delta, \sigma^2 I)$, and $M : \mathbb{R}^d \to \{0, 1\}$ be a random or deterministic function. Then, we have the following:*

*(1) If $Z = \{\mathbf{z} \in \mathbb{R}^d : \delta^T \mathbf{z} \le \beta\}$ for some $\beta$ and $Pr(M(\mathbf{X}) = 1) \ge Pr(\mathbf{X} \in Z)$, then $Pr(M(\mathbf{Y}) = 1) \ge Pr(\mathbf{Y} \in Z)$*

*(2) If $Z = \{\mathbf{z} \in \mathbb{R}^d : \delta^T \mathbf{z} \ge \beta\}$ for some $\beta$ and $Pr(M(\mathbf{X}) = 1) \le Pr(\mathbf{X} \in Z)$, then $Pr(M(\mathbf{Y}) = 1) \le Pr(\mathbf{Y} \in Z)$*

Moreover, we have the following lemma from Cohen et al. (2019).

**Lemma 2.** *Given an example $\mathbf{x}$, a number $q \in [0, 1]$, and regions $\mathcal{A}$ and $\mathcal{B}$ defined as follows:*

$$\mathcal{A} = \{\mathbf{z} : \delta^T (\mathbf{z} - \mathbf{x}) \le \sigma \|\delta\|_2 \Phi^{-1}(q)\} \tag{12}$$

$$\mathcal{B} = \{\mathbf{z} : \delta^T (\mathbf{z} - \mathbf{x}) \ge \sigma \|\delta\|_2 \Phi^{-1}(1 - q)\} \tag{13}$$

*Then, we have the following equations:*

$$Pr(\mathbf{X} \in \mathcal{A}) = q \tag{14}$$

$$Pr(\mathbf{X} \in \mathcal{B}) = q \tag{15}$$

$$Pr(\mathbf{Y} \in \mathcal{A}) = \Phi(\Phi^{-1}(q) - \frac{\|\delta\|_2}{\sigma}) \tag{16}$$

$$Pr(\mathbf{Y} \in \mathcal{B}) = \Phi(\Phi^{-1}(q) + \frac{\|\delta\|_2}{\sigma}) \tag{17}$$

*Proof.* Please refer to Cohen et al. (2019). □

Based on Lemma 1 and 2, we derive the following lemma:

**Lemma 3.** *Suppose we have an arbitrary base classifier $f$, an example $\mathbf{x}$, a set of labels which are denoted as $S$, two probabilities $\underline{p}_S$ and $\overline{p}_S$ that satisfy $\underline{p}_S \le p_S = Pr(f(\mathbf{X}) \in S) \le \overline{p}_S$, and regions $\mathcal{A}_S$ and $\mathcal{B}_S$ defined as follows:*

$$\mathcal{A}_S = \{\mathbf{z} : \delta^T (\mathbf{z} - \mathbf{x}) \le \sigma \|\delta\|_2 \Phi^{-1}(\underline{p}_S)\} \tag{18}$$

$$\mathcal{B}_S = \{\mathbf{z} : \delta^T (\mathbf{z} - \mathbf{x}) \ge \sigma \|\delta\|_2 \Phi^{-1}(1 - \overline{p}_S)\} \tag{19}$$

*Then, we have:*

$$Pr(\mathbf{X} \in \mathcal{A}_S) \le Pr(f(\mathbf{X}) \in S) \le Pr(\mathbf{X} \in \mathcal{B}_S) \tag{20}$$

$$Pr(\mathbf{Y} \in \mathcal{A}_S) \le Pr(f(\mathbf{Y}) \in S) \le Pr(\mathbf{Y} \in \mathcal{B}_S) \tag{21}$$

*Proof.* We know that $\Pr(\mathbf{X} \in \mathcal{A}_S) = \underline{p}_S$ based on Lemma 2. Combined with the condition that $\underline{p}_S \leq \Pr(f(\mathbf{X}) \in S)$, we obtain the first inequality in (20). Similarly, we can obtain the second inequality in (20). We define $M(\mathbf{z}) = \mathbb{I}(f(\mathbf{z}) \in S)$. Based on the first inequality in (20) and Lemma 1, we have the following:

$$\Pr(\mathbf{Y} \in \mathcal{A}_S) \leq \Pr(M(\mathbf{Y}) = 1) = \Pr(f(\mathbf{Y}) \in S), \tag{22}$$

which is the first inequality in (21). The second inequality in (21) can be obtained similarly. $\qquad\square$

Next, we restate Theorem 1 and show our proof.

**Theorem 1** (Certified Radius for Top-$k$ Predictions). *Suppose we are given an example* $\mathbf{x}$*, an arbitrary base classifier* $f$*,* $\epsilon \sim \mathcal{N}(0, \sigma^2 I)$*, a smoothed classifier* $g$*, an arbitrary label* $l \in \{1, 2, \cdots, c\}$*, and* $\underline{p}_l, \overline{p}_1, \cdots, \overline{p}_{l-1}, \overline{p}_{l+1}, \cdots, \overline{p}_c \in [0, 1]$ *that satisfy the following conditions:*

$$Pr(f(\mathbf{x} + \epsilon) = l) \geq \underline{p}_l \text{ and } Pr(f(\mathbf{x} + \epsilon) = i) \leq \overline{p}_i, \forall i \neq l, \tag{1}$$

*where* $\underline{p}$ *and* $\overline{p}$ *indicate lower and upper bounds of p, respectively. Let* $\overline{p}_{b_k} \geq \overline{p}_{b_{k-1}} \geq \cdots \geq \overline{p}_{b_1}$ *be the* $k$ *largest ones among* $\{\overline{p}_1, \cdots, \overline{p}_{l-1}, \overline{p}_{l+1}, \cdots, \overline{p}_c\}$*, where ties are broken uniformly at random. Moreover, we denote by* $S_t = \{b_1, b_2, \cdots, b_t\}$ *the set of* $t$ *labels with the smallest probability upper bounds in the* $k$ *largest ones and by* $\overline{p}_{S_t} = \sum_{j=1}^{t} \overline{p}_{b_j}$ *the sum of the* $t$ *probability upper bounds, where* $t = 1, 2, \cdots, k$*. Then, we have:*

$$l \in g_k(\mathbf{x} + \delta), \forall ||\delta||_2 < R_l, \tag{2}$$

*where* $R_l$ *is the unique solution to the following equation:*

$$\Phi(\Phi^{-1}(\underline{p}_l) - \frac{R_l}{\sigma})) - \min_{t=1}^{k} \frac{\Phi(\Phi^{-1}(\overline{p}_{S_t}) + \frac{R_l}{\sigma}))}{t} = 0, \tag{3}$$

*where* $\Phi$ *and* $\Phi^{-1}$ *are the cumulative distribution function and its inverse of the standard Gaussian distribution, respectively.*

*Proof.* Roughly speaking, our idea is to make the probability that the base classifier $f$ predicts $l$ when taking $\mathbf{Y}$ as input larger than the smallest one among the probabilities that $f$ predicts for a set of arbitrary $k$ labels selected from all labels except $l$. For simplicity, we let $\Gamma = \{1, 2, \cdots, c\} \setminus \{l\}$, i.e., all labels except $l$. We denote by $\Gamma_k$ a set of $k$ labels in $\Gamma$. We aim to find a certified radius $R_l$ such that we have $\max_{\Gamma_k \subseteq \Gamma} \min_{i \in \Gamma_k} \Pr(f(\mathbf{Y}) = i) < \Pr(f(\mathbf{Y}) = l)$, which guarantees $l \in g_k(\mathbf{x} + \delta)$. We first upper bound the minimal probability $\min_{i \in \Gamma_k} \Pr(f(\mathbf{Y}) = i)$ for a given $\Gamma_k$, and then we upper bound the maximum value of the minimal probability among all possible $\Gamma_k \subseteq \Gamma$. Finally, we obtain the certified radius $R_l$ via letting the upper bound of the maximum value smaller than $\Pr(f(\mathbf{Y}) = l)$.

**Bounding** $\min_{i \in \Gamma_k} \Pr(f(\mathbf{Y}) = i)$ **for a given** $\Gamma_k$**:** We use $S$ to denote a non-empty subset of $\Gamma_k$ and use $|S|$ to denote its size. We define $\overline{p}_S = \sum_{i \in S} \overline{p}_i$, which is the sum of the upper bounds of the probabilities for the labels in $S$. Moreover, we define the following region associated with the set $S$:

$$\mathcal{B}_S = \{\mathbf{z} : \delta^T(\mathbf{z} - \mathbf{x}) \geq \sigma \|\delta\|_2 \Phi^{-1}(1 - \overline{p}_S)\} \tag{23}$$

We have $\Pr(f(\mathbf{Y}) \in S) \leq \Pr(\mathbf{Y} \in \mathcal{B}_S)$ by applying Lemma 3 to the set $S$. In addition, we have $\sum_{i \in S} \Pr(f(\mathbf{Y}) = i) = \Pr(f(\mathbf{Y}) \in S)$. Therefore, we have:

$$\sum_{i \in S} \Pr(f(\mathbf{Y}) = i) = \Pr(f(\mathbf{Y}) \in S) \leq \Pr(\mathbf{Y} \in \mathcal{B}_S) \tag{24}$$

Moreover, we have:

$$\min_{i \in \Gamma_k} \Pr(f(\mathbf{Y}) = i) \leq \min_{i \in S} \Pr(f(\mathbf{Y}) = i) \tag{25}$$

$$\leq \frac{\sum_{i \in S} \Pr(f(\mathbf{Y}) = i)}{|S|} \tag{26}$$

$$\leq \frac{\Pr(\mathbf{Y} \in \mathcal{B}_S)}{|S|}, \tag{27}$$

where we have the first inequality because $S$ is a subset of $\Gamma_k$ and we have the second inequality because the smallest value in a set is no larger than the average value of the set. Equation 27 holds for any $S \subseteq \Gamma_k$. Therefore, by taking all possible sets $S$ into consideration, we have the following:

$$\min_{i \in \Gamma_k} \Pr(f(\mathbf{Y}) = i) \leq \min_{S \subseteq \Gamma_k} \frac{\Pr(\mathbf{Y} \in \mathcal{B}_S)}{|S|} \tag{28}$$

$$= \min_{t=1}^{k} \min_{S \subseteq \Gamma_k, |S|=t} \frac{\Pr(\mathbf{Y} \in \mathcal{B}_S)}{t} \tag{29}$$

$$= \min_{t=1}^{k} \frac{\Pr(\mathbf{Y} \in \mathcal{B}_{S_t})}{t}, \tag{30}$$

where $S_t$ is the set of $t$ labels in $\Gamma_k$ whose probability upper bounds are the smallest, where ties are broken uniformly at random. We have Equation 30 from Equation 29 because $\Pr(\mathbf{Y} \in \mathcal{B}_S)$ decreases as $\overline{p}_S$ decreases.

**Bounding $\max_{\Gamma_k \subseteq \Gamma} \min_{i \in \Gamma_k} \Pr(f(\mathbf{Y}) = i)$:** Since $\Pr(\mathbf{Y} \in \mathcal{B}_{S_t})$ increases as $\overline{p}_{S_t}$ increases, Equation 30 reaches its maximum value when $\Gamma_k = \{b_1, b_2, \cdots, b_k\}$, i.e., when $\Gamma_k$ is the set of $k$ labels in $\Gamma$ with the largest probability upper bounds. Formally, we have:

$$\max_{\Gamma_k \subseteq \Gamma} \min_{i \in \Gamma_k} \Pr(f(\mathbf{Y}) = i) \leq \min_{t=1}^{k} \frac{\Pr(\mathbf{Y} \in \mathcal{B}_{S_t})}{t}, \tag{31}$$

where $S_t = \{b_1, b_2, \cdots, b_t\}$.

**Obtaining $R_l$:** According to Lemma 3, we have the following for $S = \{l\}$:

$$\Pr(f(\mathbf{Y}) = l) \geq \Pr(\mathbf{Y} \in \mathcal{A}_{\{l\}}) \tag{32}$$

Recall that our goal is to make $\Pr(f(\mathbf{Y}) = l) > \max_{\Gamma_k \subseteq \Gamma} \min_{i \in \Gamma_k} \Pr(f(\mathbf{Y}) = i)$. It suffices to let:

$$\Pr(\mathbf{Y} \in \mathcal{A}_{\{l\}}) > \min_{t=1}^{k} \frac{\Pr(\mathbf{Y} \in \mathcal{B}_{S_t})}{t}. \tag{33}$$

According to Lemma 2, we have $\Pr(\mathbf{Y} \in \mathcal{A}_{\{l\}}) = \Phi(\Phi^{-1}(\underline{p_l}) - \frac{||\delta||_2}{\sigma}))$ and $\Pr(\mathbf{Y} \in \mathcal{B}_{S_t}) = \Phi(\Phi^{-1}(\overline{p}_{S_t}) + \frac{||\delta||_2}{\sigma}))$. Therefore, we have the following constraint on $\delta$:

$$\Phi(\Phi^{-1}(\underline{p_l}) - \frac{||\delta||_2}{\sigma})) - \min_{t=1}^{k} \frac{\Phi(\Phi^{-1}(\overline{p}_{S_t}) + \frac{||\delta||_2}{\sigma}))}{t} > 0. \tag{34}$$

Since the left-hand side of the above inequality 1) decreases as $||\delta||_2$ increases, 2) is larger than 0 when $||\delta||_2 \rightarrow -\infty$, and 3) is smaller than 0 when $||\delta||_2 \rightarrow \infty$, we have the constraint $||\delta||_2 < R_l$, where $R_l$ is the unique solution to the following equation:

$$\Phi(\Phi^{-1}(\underline{p_l}) - \frac{R_l}{\sigma})) - \min_{t=1}^{k} \frac{\Phi(\Phi^{-1}(\overline{p}_{S_t}) + \frac{R_l}{\sigma}))}{t} = 0. \tag{35}$$

$\square$

## B    Proof of Theorem 2

Following the terminology we used in proving Theorem 1, we define a region $\mathcal{A}_{\{l\}}$ as follows:

$$\mathcal{A}_{\{l\}} = \{\mathbf{z} : \delta^T(\mathbf{z} - \mathbf{x}) \leq \sigma \, ||\delta||_2 \, \Phi^{-1}(\underline{p_l})\}. \tag{36}$$

According to Lemma 2, we have $\Pr(\mathbf{X} \in \mathcal{A}_{\{l\}}) = \underline{p_l}$. We first show the following lemma, which is the key to prove our Theorem 2.

**Lemma 4.** *Assuming we have $\underline{p_l} + \sum_{j=1}^{k} \overline{p}_{b_j} \leq 1$. For any perturbation $||\delta||_2 > R_l$, there exists $k$ disjoint regions $\mathcal{C}_{b_j} \subseteq \mathbb{R}^d \setminus \mathcal{A}_{\{l\}}, j \in \{1, 2, \cdots, k\}$ that satisfy the following:*

$$Pr(\mathbf{X} \in \mathcal{C}_{b_j}) = \overline{p}_{b_j}, \, \forall j \in \{1, 2, \cdots, k\} \tag{37}$$

$$Pr(\mathbf{Y} \in \mathcal{C}_{b_j}) \geq \min_{t=1}^{k} \frac{Pr(\mathbf{Y} \in \mathcal{B}_{S_t})}{t}, \, \forall j \in \{1, 2, \cdots, k\}, \tag{38}$$

*where the random variables $\mathbf{X}$ and $\mathbf{Y}$ are defined in Equation 10 and 11, respectively; and $\{b_1, b_2, \cdots, b_k\}$ and $S_t$ are defined in Theorem 1.*

*Proof.* Our proof is based on *mathematical induction* and the *intermediate value theorem*. For convenience, we defer the proof to Appendix B.1. □

Next, we restate Theorem 2 and show our proof.

**Theorem 2** (Tightness of the Certified Radius). *Assuming we have $\underline{p_l} + \sum_{j=1}^{k} \overline{p}_{b_j} \leq 1$ and $\underline{p_l} + \sum_{i=1,\cdots,l-1,l+1,\cdots,c} \overline{p}_i \geq 1$. Then, for any perturbation $||\delta||_2 > R_l$, there exists a base classifier $f^*$ consistent with (1) but we have $l \notin g_k(\mathbf{x} + \delta)$.*

*Proof.* Our idea is to construct a base classifier such that $l$ is not among the top-$k$ labels predicted by the smoothed classifier for any perturbed example $\mathbf{x} + \delta$ when $||\delta||_2 > R_l$. First, according to Lemma 4, we know there exists $k$ disjoint regions $\mathcal{C}_{b_j} \subseteq \mathbb{R}^d \setminus \mathcal{A}_{\{l\}}$, $j \in \{1, 2, \cdots, k\}$ that satisfy Equation 37 and 38. Moreover, we divide the remaining region $\mathbb{R}^d \setminus (\mathcal{A}_{\{l\}} \cup_{j=1}^{k} \mathcal{C}_{b_j})$ into $c-k-1$ regions, which we denote as $\mathcal{C}_{b_{k+1}}, \mathcal{C}_{b_{k+2}}, \cdots, \mathcal{C}_{b_{c-1}}$ and satisfy $\Pr(\mathbf{X} \in \mathcal{C}_{b_j}) \leq \overline{p}_{b_j}$ for $j \in \{k+1, k+2, \cdots, c-1\}$. Note that $b_1, b_2, \cdots, b_{c-1}$ is some permutation of $\{1, 2, \cdots, c\} \setminus \{l\}$. We can divide the remaining region into such $c-k-1$ regions because $\underline{p_l} + \sum_{i=1,\cdots,l-1,l+1,\cdots,c} \overline{p}_i \geq 1$. Then, based on these regions, we construct the following base classifier:

$$f^*(\mathbf{x}) = \begin{cases} l, & \text{if } \mathbf{x} \in \mathcal{A}_{\{l\}} \\ b_j, & \text{if } \mathbf{x} \in \mathcal{C}_{b_j}, \forall j \in \{1, 2, \cdots, c-1\} \end{cases} \tag{39}$$

Based on the definition of $f^*$, we have the following:

$$\Pr(f^*(\mathbf{X}) = l) = \Pr(\mathbf{X} \in \mathcal{A}_{\{l\}}) = \underline{p_l} \tag{40}$$

$$\Pr(f^*(\mathbf{X}) = b_j) = \Pr(\mathbf{X} \in \mathcal{C}_{b_j}) = \overline{p}_{b_j}, \; j = 1, 2, \cdots, k \tag{41}$$

$$\Pr(f^*(\mathbf{X}) = b_j) = \Pr(\mathbf{X} \in \mathcal{C}_{b_j}) \leq \overline{p}_{b_j}, \; j = k+1, k+2, \cdots, c-1 \tag{42}$$

Therefore, $f^*$ satisfies the conditions in (1). Next, we show that $l$ is not among the top-$k$ labels predicted by the smoothed classifier for any perturbed example $\mathbf{x} + \delta$ when $||\delta||_2 > R_l$. Specifically, we have:

$$\Pr(f^*(\mathbf{Y}) = l | \, ||\delta||_2 > R_l) \tag{43}$$

$$= \Pr(\mathbf{Y} \in \mathcal{A}_{\{l\}} | \, ||\delta||_2 > R_l) \qquad (\text{Definition of } f^*) \tag{44}$$

$$< \Pr(\mathbf{Y} \in \mathcal{A}_{\{l\}} | \, ||\delta||_2 = R_l) \qquad (\Pr(\mathbf{Y} \in \mathcal{A}_{\{l\}}) \text{ increases as } ||\delta||_2 \text{ decreases}) \tag{45}$$

$$= \min_{t=1}^{k} \frac{\Pr(\mathbf{Y} \in \mathcal{B}_{S_t} | \, ||\delta||_2 = R_l)}{t} \qquad (\text{Definition of } R_l) \tag{46}$$

$$< \min_{t=1}^{k} \frac{\Pr(\mathbf{Y} \in \mathcal{B}_{S_t} | \, ||\delta||_2 > R_l)}{t} \qquad (\Pr(\mathbf{Y} \in \mathcal{B}_{S_t}) \text{ increases as } ||\delta||_2 \text{ increases}) \tag{47}$$

$$\leq \Pr(\mathbf{Y} \in \mathcal{C}_{b_j} | \, ||\delta||_2 > R_l) \qquad (Lemma \; 4) \tag{48}$$

$$= \Pr(f^*(\mathbf{Y}) = b_j | \, ||\delta||_2 > R_l) \qquad (\text{Definition of } f^*), \tag{49}$$

where $j = 1, 2, \cdots, k$. Since we have found $k$ labels whose probabilities are larger than the probability of the label $l$, we have $l \notin g_k(\mathbf{x} + \delta)$ when $||\delta||_2 > R_l$. □

## B.1 PROOF OF LEMMA 4

We first define some key notations and lemmas that will be used in our proof.

**Definition 1** ($\mathcal{C}(q_1, q_2), \mathcal{C}'(q_1, q_2), r_x(q_1, q_2), r_y(q_1, q_2)$). *Given two values $q_1$ and $q_2$ that satisfy $0 \leq q_1 < q_2 \leq 1$, we define the following region:*

$$\mathcal{C}(q_1, q_2) = \{\mathbf{z} : \sigma \, ||\delta||_2 \, \Phi^{-1}(1 - q_2) < \delta^T(\mathbf{z} - \mathbf{x}) \leq \sigma \, ||\delta||_2 \, \Phi^{-1}(1 - q_1)\} \tag{50}$$

*According to Lemma 2, we have:*

$$Pr(\mathbf{X} \in \mathcal{C}(q_1, q_2)) = q_2 - q_1, \tag{51}$$

*where the Gaussian random variable $\mathbf{X}$ is defined in Equation 10. Moreover, assuming we have pairs of $(q_1^i, q_2^i)$, $i = 1, 2, 3, \cdots$, where $q_1^i, q_2^i \in [q_1, q_2]$, $\forall i$. We define the following region:*

$$\mathcal{C}'(q_1, q_2) = \mathcal{C}(q_1, q_2) \setminus (\cup_i \mathcal{C}(q_1^i, q_2^i)). \tag{52}$$

$\mathcal{C}'(q_1, q_2)$ is the remaining region of $\mathcal{C}(q_1, q_2)$ excluding $\mathcal{C}(q_1^1, q_2^1), \mathcal{C}(q_1^2, q_2^2), \cdots$. Given two values $q_1^\lambda$ and $q_2^\lambda$ that satisfy $q_1 \leq q_1^\lambda \leq q_2^\lambda \leq q_2$, we also define the following two functions:

$$r_x(q_1^\lambda, q_2^\lambda) = Pr(\mathbf{X} \in \mathcal{C}'(q_1, q_2) \cap \mathcal{C}(q_1^\lambda, q_2^\lambda)) \tag{53}$$

$$r_y(q_1^\lambda, q_2^\lambda) = Pr(\mathbf{Y} \in \mathcal{C}'(q_1, q_2) \cap \mathcal{C}(q_1^\lambda, q_2^\lambda)), \tag{54}$$

where the random variables $\mathbf{X}$ and $\mathbf{Y}$ are defined in Equation 10 and 11, respectively.

Next, we show a key property of our defined functions $r_x(q_1^\lambda, q_2^\lambda)$ and $r_y(q_1^\lambda, q_2^\lambda)$.

**Lemma 5.** If $r_x(q_1^\kappa, q_2^\kappa) \leq r_x(q_1^\lambda, q_2^\lambda)$ and $q_2^\kappa \geq q_2^\lambda$ (or $q_1^\kappa \geq q_1^\lambda$), then we have $r_y(q_1^\kappa, q_2^\kappa) \leq r_y(q_1^\lambda, q_2^\lambda)$.

*Proof.* We consider three scenarios.

**Scenario I:** $q_2^\kappa \geq q_1^\kappa \geq q_2^\lambda \geq q_1^\lambda$. We denote $h_x$ and $h_y$ as the probability densities for the random variables $\mathbf{X}$ and $\mathbf{Y}$, respectively. Then, we have $h_x(\mathbf{z}) = (\frac{1}{\sqrt{2\pi}\sigma})^d \exp(-\frac{\sum_{i=1}^d (z_i - x_i)^2}{2\sigma^2})$ and $h_y(\mathbf{z}) = (\frac{1}{\sqrt{2\pi}\sigma})^d \exp(-\frac{\sum_{i=1}^d (z_i - x_i - \delta_i)^2}{2\sigma^2})$. Therefore, the ratio of the probability density of $\mathbf{Y}$ and the probability density of $\mathbf{X}$ at a given point $\mathbf{z}$ is as follows:

$$\frac{h_y(\mathbf{z})}{h_x(\mathbf{z})} = \exp(\frac{\delta^T(\mathbf{z} - \mathbf{x})}{\sigma^2} - \frac{\|\delta\|_2}{2\sigma^2}) \tag{55}$$

Next, we compare the ratio for the points in different regions and have the following:

$$\frac{h_y(\mathbf{z}|\mathbf{z} \in \mathcal{C}'(q_1^\kappa, q_2^\kappa))}{h_x(\mathbf{z}|\mathbf{z} \in \mathcal{C}'(q_1^\kappa, q_2^\kappa))} \tag{56}$$

$$\leq \exp(\frac{\sigma \|\delta\|_2 \Phi^{-1}(1 - q_1^\kappa)}{\sigma^2} - \frac{\|\delta\|_2}{2\sigma^2}) \tag{57}$$

$$\leq \exp(\frac{\sigma \|\delta\|_2 \Phi^{-1}(1 - q_2^\lambda)}{\sigma^2} - \frac{\|\delta\|_2}{2\sigma^2}) \tag{58}$$

$$\leq \frac{h_y(\mathbf{z}|\mathbf{z} \in \mathcal{C}'(q_1^\lambda, q_2^\lambda))}{h_x(\mathbf{z}|\mathbf{z} \in \mathcal{C}'(q_1', q_2^\lambda))} \tag{59}$$

The Equation 57 from 56 is based on Equation 55 and the fact that $\delta^T(\mathbf{z} - \mathbf{x}) \leq \sigma \|\delta\|_2 \Phi^{-1}(1 - q_1^\kappa)$ for any point $\mathbf{z}$ in the region $\mathcal{C}'(q_1^\kappa, q_2^\kappa)$ from Definition 1. Similarly, we can obtain Equation 59 from 58. We note that the Equation 58 from 57 is because $q_2^\lambda \leq q_1^\kappa$. Based on Equation 57 and 58, we know that there exists a real number $u$ such that:

$$\exp(\frac{\sigma \|\delta\|_2 \Phi^{-1}(1 - q_1^\kappa)}{\sigma^2} - \frac{\|\delta\|_2}{2\sigma^2}) \leq u \leq \exp(\frac{\sigma \|\delta\|_2 \Phi^{-1}(1 - q_2^\lambda)}{\sigma^2} - \frac{\|\delta\|_2}{2\sigma^2}) \tag{60}$$

Combining the Equation 56, 57, and 60, we have the following:

$$h_y(\mathbf{z}|\mathbf{z} \in \mathcal{C}'(q_1^\kappa, q_2^\kappa)) \leq u \cdot h_x(\mathbf{z}|\mathbf{z} \in \mathcal{C}'(q_1^\kappa, q_2^\kappa)) \tag{61}$$

Taking an integral on both sides of the Equation 61 in the region $\mathcal{C}'(q_1^\kappa, q_2^\kappa)$ and recalling the definition of $r_x(q_1^\kappa, q_2^\kappa)$ and $r_y(q_1^\kappa, q_2^\kappa)$, we have the following:

$$r_y(q_1^\kappa, q_2^\kappa) \leq u \cdot r_x(q_1^\kappa, q_2^\kappa) \tag{62}$$

Similarly, we have:

$$u \cdot r_x(q_1^\lambda, q_2^\lambda) \leq r_y(q_1^\lambda, q_2^\lambda) \tag{63}$$

Based on Equation 62, 63, and the condition that $r_x(q_1^\kappa, q_2^\kappa) \leq r_x(q_1^\lambda, q_2^\lambda)$, we have the following:

$$r_y(q_1^\kappa, q_2^\kappa) \leq r_y(q_1^\lambda, q_2^\lambda) \tag{64}$$

**Scenario II:** $q_2^\kappa \geq q_2^\lambda \geq q_1^\kappa \geq q_1^\lambda$. We have:

$$r_x(q_1^\kappa, q_2^\kappa) = r_x(q_1^\kappa, q_2^\lambda) + r_x(q_2^\lambda, q_2^\kappa) \leq r_x(q_1^\lambda, q_1^\kappa) + r_x(q_1^\kappa, q_2^\lambda) = r_x(q_1^\lambda, q_2^\lambda) \tag{65}$$

Therefore, we have the following equation:

$$r_x(q_2^\lambda, q_2^\kappa) \leq r_x(q_1^\lambda, q_1^\kappa) \tag{66}$$

Similar to **Scenario I**, we know that there exists $u$ such that:

$$\frac{h_y(\mathbf{z}|\mathbf{z} \in \mathcal{C}'(q_2^\lambda, q_2^\kappa))}{h_x(\mathbf{z}|\mathbf{z} \in \mathcal{C}'(q_2^\lambda, q_2^\kappa))} \leq u \leq \frac{h_y(\mathbf{z}|\mathbf{z} \in \mathcal{C}'(q_1^\lambda, q_1^\kappa))}{h_x(\mathbf{z}|\mathbf{z} \in \mathcal{C}'(q_1^\lambda, q_1^\kappa))} \tag{67}$$

Similar to **Scenario I**, we have the following based on Equation 66:

$$r_y(q_2^\lambda, q_2^\kappa) \leq r_y(q_1^\lambda, q_1^\kappa) \tag{68}$$

Therefore, we have the following:

$$r_y(q_1^\kappa, q_2^\kappa) = r_y(q_1^\kappa, q_2^\lambda) + r_y(q_2^\lambda, q_2^\kappa) \leq r_y(q_1^\lambda, q_1^\kappa) + r_y(q_1^\kappa, q_2^\lambda) = r_y(q_1^\lambda, q_2^\lambda) \tag{69}$$

**Scenario III:** $q_2^\kappa \geq q_2^\lambda \geq q_1^\lambda \geq q_1^\kappa$. As $r_x(q_1^\kappa, q_2^\kappa) \leq r_x(q_1^\lambda, q_2^\lambda)$, we have $r_x(q_1^\kappa, q_2^\kappa) = r_x(q_1^\lambda, q_2^\lambda)$. Therefore, we have $r_y(q_1^\kappa, q_2^\kappa) = r_y(q_1^\lambda, q_2^\lambda)$. $\square$

Next, we list the well-known *Intermediate Value Theorem* and show several other properties of our defined functions $r_x(q_1^\lambda, q_2^\lambda)$ and $r_y(q_1^\lambda, q_2^\lambda)$.

**Lemma 6** (Intermediate Value Theorem). *If a function $F$ is continuous at every point in the interval $[a, b]$ and $(F(a) - v) \cdot (F(b) - v) \leq 0$, then there exists $x$ such that $F(x) = v$.*

Roughly speaking, the Intermediate Value Theorem tells us that if a continuous function has values no larger and no smaller (or no smaller and no larger) than $v$ at the two end points of an interval, respectively, then the function takes value $v$ at some point in the interval.

**Lemma 7.** *Given two probabilities $q_x, q_y$, if we have:*

$$q_x \leq r_x(q_1, q_2) \tag{70}$$

*Then, there exists $q_1', q_2' \in [q_1, q_2]$ such that:*

$$r_x(q_1, q_2') = q_x \tag{71}$$

$$r_x(q_1', q_2) = q_x \tag{72}$$

*Furthermore, if we have:*

$$(r_y(q_1, q_2') - q_y) \cdot (r_y(q_1', q_2) - q_y) \leq 0, \tag{73}$$

*then there exists $q_1'', q_2'' \in [q_1, q_2]$ such that:*

$$r_x(q_1'', q_2'') = q_x \tag{74}$$

$$r_y(q_1'', q_2'') = q_y \tag{75}$$

*Proof.* We define function $F(x) = r_x(q_1, x)$. Then, we have $(F(q_1) - q_x) \cdot (F(q_2) - q_x) \leq 0$ since $F(q_1) = 0 \leq q_x$ and $F(q_2) = r_x(q_1, q_2) > q_x$ based on Equation 70. Therefore, according to Lemma 6, there exists $q_1' \in [q_1, q_2]$ such that:

$$r_x(q_1, q_2') = q_x \tag{76}$$

Similarly, we can prove that there exists $q_2' \in [q_1, q_2]$ such that $r_x(q_1', q_2) = q_x$.

For any $q_2^e \in [q_2', q_2]$, we define $H(x) = r_x(x, q_2^e)$. Then, we know $H(q_1) = r_x(q_1, q_2^e) \geq r_x(q_1, q_2') = q_x$ since $q_2^e \geq q_2'$. Moreover, we have $H(q_2^e) = 0 \leq q_x$. Therefore, we have $(H(q_1) - q_x) \cdot (H(q_2^e) - q_x) \leq 0$. According to Lemma 6, we know there exists $q_1^e \in [q_1, q_2^e]$ such that $r_x(q_1^e, q_2^e) = q_x$ for arbitrary $q_2^e \in [q_2', q_2]$. We define $G(x) = r_y(q_1^e, x)$ where $x \in [q_2', q_2]$, and $q_1^e$ are a value such that $r_x(q_1^e, x) = q_x$ for a given $x$. When $x = q_2'$, we can let $q_1^e = q_1$ since $r_x(q_1, q_2') = q_x$, and when $x = q_1$, we can let $q_1^e = q_2'$ since $r_x(q_2', q_2) = q_x$. Based on Equation 73 and Lemma 6, we know that there exists $x \in [q_2', q_2]$ such that $G(x) = q_y$. Therefore, there exists $q_1''$ and $q_2''$ such that:

$$r_x(q_1'', q_2'') = q_x \tag{77}$$

$$r_y(q_1'', q_2'') = q_y \tag{78}$$

$\square$

**Lemma 8.** *Assuming we have $q_1^\lambda = 0$, $q_2^\lambda = \overline{p}_{S_t}$. If $q_2^\lambda = \overline{p}_{S_t} \leq \min_i q_1^i$, then we have the following:*

$$r_x(q_1^\lambda, q_2^\lambda) = q_2^\lambda - q_1^\lambda = \overline{p}_{S_t} \tag{79}$$

$$r_y(q_1^\lambda, q_2^\lambda) = Pr(\mathbf{Y} \in \mathcal{B}_{S_t}) \tag{80}$$

*Proof.* If $q_2^\lambda \leq \min_i q_1^i$, then we have $\mathcal{C}'(q_1, q_2) \cap \mathcal{C}(q_1^\lambda, q_2^\lambda) = \mathcal{C}(q_1^\lambda, q_2^\lambda)$ since no region is excluded. Therefore, we have $r_y(q_1^\lambda, q_2^\lambda) = \Pr(\mathbf{Y} \in \mathcal{C}'(q_1, q_2) \cap \mathcal{C}(q_1^\lambda, q_2^\lambda)) = \Pr(\mathbf{Y} \in \mathcal{C}(q_1^\lambda, q_2^\lambda)) = q_2^\lambda - q_1^\lambda = \overline{p}_{S_t}$ based on Equation 51. We note that $\mathcal{C}(q_1^\lambda, q_2^\lambda) = \mathcal{B}_{S_t}$ when $q_1^\lambda = 0$ and $q_2^\lambda = \overline{p}_{S_t}$. Therefore, we can obtain Equation 80 based on the definition of $r_y(q_1^\lambda, q_2^\lambda)$ from Definition 1. $\qquad\square$

**Lemma 9.** *If we have $q_1 \leq q_2^o \leq q_2$, then we have the following:*

$$r_y(q_1, q_2^o) \geq \frac{r_y(q_1, q_2)}{\lceil r_x(q_1, q_2)/r_x(q_1, q_2^o) \rceil} \tag{81}$$

*Proof.* By applying Lemma 5. $\qquad\square$

We further generalize Lemma 5 to two regions. Specifically, we have the following lemma:

**Lemma 10.** *Assuming we have a region $\mathcal{C}_w \subseteq \mathcal{C}(q_1^w, q_2^w)$ and we have $\mathcal{C}'(q_1, q_2) \cap \mathcal{C}(q_1^w, q_2^w) = \emptyset$. If $q_1 \geq q_1^w$, $q_2 \geq q_2^w$ and $r_x(q_1', q_2') \leq Pr(\mathbf{X} \in \mathcal{C}_w)$, then we have:*

$$r_y(q_1, q_2) \leq Pr(\mathbf{Y} \in \mathcal{C}_w) \tag{82}$$

*Proof.* We let $q_1 = \max(q_1, q_2^w)$. As $\mathcal{C}'(q_1, q_2) \cap \mathcal{C}(q_1^w, q_2^w) = \emptyset$ and $q_1 \geq q_1^w$, we can obtain the conclusion by applying Lemma 5 on $\mathcal{C}'(q_1, q_2) \cup \mathcal{C}_w$. $\qquad\square$

Next, we restate Lemma 4 and show our proof.

**Lemma 4.** *Assuming we have $\underline{p_l} + \sum_{j=1}^k \overline{p}_{b_j} \leq 1$. For any perturbation $\|\delta\|_2 > R_l$, there exists $k$ disjoint regions $\mathcal{C}_{b_j} \subseteq \mathbb{R}^d \setminus \mathcal{A}_{\{l\}}$, $j \in \{1, 2, \cdots, k\}$ that satisfy the following:*

$$Pr(\mathbf{X} \in \mathcal{C}_{b_j}) = \overline{p}_{b_j}, \ \forall j \in \{1, 2, \cdots, k\} \tag{37}$$

$$Pr(\mathbf{Y} \in \mathcal{C}_{b_j}) \geq \min_{t=1}^k \frac{Pr(\mathbf{Y} \in \mathcal{B}_{S_t})}{t}, \ \forall j \in \{1, 2, \cdots, k\}, \tag{38}$$

*where the random variables $\mathbf{X}$ and $\mathbf{Y}$ are defined in Equation 10 and 11, respectively; and $\{b_1, b_2, \cdots, b_k\}$ and $S_t$ are defined in Theorem 1.*

*Proof.* Our proof leverages *Mathematical Induction*, which contains two steps. In the first step, we show that the statement holds initially. In the second step, we show that if the statement is true for the $m$th iteration, then it also holds for the $(m+1)$th iteration. Without loss of generality, we assume $\tau = \mathrm{argmin}_{t=1}^k \frac{\Pr(\mathbf{Y} \in \mathcal{B}_{S_t})}{t}$. Therefore, we have the following:

$$\forall i \neq \tau, \frac{\Pr(\mathbf{Y} \in \mathcal{B}_{S_i})}{i} \geq \frac{\Pr(\mathbf{Y} \in \mathcal{B}_{S_\tau})}{\tau} \tag{83}$$

Recall the definition of $\mathcal{B}_{S_\tau}$ and we have the following:

$$\mathcal{B}_{S_\tau} = \{\mathbf{z} : \delta^T(\mathbf{z} - \mathbf{x}) \geq \sigma \|\delta\|_2 \Phi^{-1}(1 - \overline{p}_{S_\tau})\}, \tag{84}$$

where $\overline{p}_{S_\tau} = \sum_{j \in S_\tau} \overline{p}_j$. We can split $\mathcal{B}_{S_k}$ into two parts: $\mathcal{B}_{S_\tau}$ and $\mathcal{B}_{S_k} \setminus \mathcal{B}_{S_\tau}$. We will show that $\forall j \in [1, \tau]$, we can find disjoint $\mathcal{C}_{b_j} \subseteq \mathcal{B}_{S_\tau}$ whose union is $\mathcal{B}_{S_\tau}$ such that:

$$\Pr(\mathbf{X} \in \mathcal{C}_{b_j}) = \overline{p}_{b_j} \tag{85}$$

$$\Pr(\mathbf{Y} \in \mathcal{C}_{b_j}) = \frac{\Pr(\mathbf{Y} \in \mathcal{B}_{S_\tau})}{\tau} \tag{86}$$

For the other part, we will show that $\forall j \in [\tau + 1, k]$, we can find disjoint $\mathcal{C}_{b_j} \subseteq \mathcal{B}_{S_k} \setminus \mathcal{B}_{S_\tau}$ whose union is $\mathcal{B}_{S_k} \setminus \mathcal{B}_{S_\tau}$ such that:

$$\Pr(\mathbf{X} \in \mathcal{C}_{b_j}) = \overline{p}_{b_j} \tag{87}$$

$$\Pr(\mathbf{Y} \in \mathcal{C}_{b_j}) \geq \frac{\Pr(\mathbf{Y} \in \mathcal{B}_{S_\tau})}{\tau} \tag{88}$$

We first show that $\forall j \in [1, \tau]$, we can find $\mathcal{C}_{b_j} \subseteq \mathcal{B}_{S_\tau}$ that satisfy Equation 85 and 86. Since our proof leverages Mathematical Induction, we iteratively construct each $\mathcal{C}_{b_j}, \forall j \in [1, \tau]$. Specifically, we first show that we can find $\mathcal{C}_{b_\tau} \subseteq \mathcal{B}_{S_\tau}$ that satisfies the requirements. Then, assuming we can find $\mathcal{C}_{b_\tau}, \cdots, \mathcal{C}_{b_{\tau-m+1}}$, we show that we can find $\mathcal{C}_{b_{\tau-m}} \subseteq \mathcal{B}_{S_\tau} \setminus (\cup_{j=\tau-m+1}^{\tau} \mathcal{C}_{b_j})$. We will leverage Lemma 7 to prove the existence for each $\mathcal{C}_{b_j}$. Next, we show the two steps.

**Step I:** We show that we can find $\mathcal{C}_{b_\tau} \subseteq \mathcal{B}_{S_\tau}$ that satisfies Equation 85 and 86. We let $q_1 = 0$ and $q_2 = \overline{p}_{S_\tau}$, and we define the following region:

$$\mathcal{C}'(q_1, q_2) = \mathcal{C}(0, \overline{p}_{S_\tau}) = \mathcal{B}_{S_\tau} \tag{89}$$

We have:

$$r_x(q_1, q_2) = \Pr(\mathbf{X} \in \mathcal{C}'(q_1, q_2)) = \overline{p}_{S_\tau} \tag{90}$$

$$r_y(q_1, q_2) = \Pr(\mathbf{Y} \in \mathcal{C}'(q_1, q_2)) = \Pr(\mathbf{Y} \in \mathcal{B}_{S_\tau}), \tag{91}$$

which can be directly obtained as $\mathcal{C}'(q_1, q_2) = \mathcal{B}_{S_\tau}$. As we have $\overline{p}_{b_\tau} \leq r_x(q_1, q_2) = \overline{p}_{S_\tau} = \sum_{j=1}^{\tau} \overline{p}_{b_j}$, there exist $q_1' = \overline{p}_{S_\tau} - \overline{p}_{b_\tau}, q_2' = \overline{p}_{b_\tau}$ such that:

$$r_x(q_1, q_2') = \overline{p}_{b_\tau} \tag{92}$$

$$r_x(q_1', q_2) = \overline{p}_{b_\tau} \tag{93}$$

Moreover, we have the following:

$$r_y(q_1, q_2') \geq r_y(q_1, \overline{p}_{b_1}) = \Pr(\mathbf{Y} \in \mathcal{C}(q_1, \overline{p}_{b_1})) = \Pr(\mathbf{Y} \in \mathcal{B}_{S_1}) \geq \frac{\Pr(\mathbf{Y} \in \mathcal{B}_{S_\tau})}{\tau} \tag{94}$$

The equality in the middle is from Lemma 8, the left inequality is because $q_2' = \overline{p}_{b_\tau} \geq \overline{p}_{b_1}$, and the right inequality is from Equation 83. Furthermore, we have the following:

$$r_y(q_1', q_2) \tag{95}$$

$$= \Pr(\mathbf{Y} \in \mathcal{C}'(q_1, q_2) \cap \mathcal{C}(q_1', q_2)) \tag{96}$$

$$= \Pr(\mathbf{Y} \in \mathcal{C}(\overline{p}_{S_\tau} - \overline{p}_{b_\tau}, \overline{p}_{S_\tau})) \tag{97}$$

$$= \Pr(\mathbf{Y} \in \mathcal{C}(0, \overline{p}_{S_\tau})) - \Pr(\mathbf{Y} \in \mathcal{C}(0, \overline{p}_{S_\tau} - \overline{p}_{b_\tau})) \tag{98}$$

$$= \Pr(\mathbf{Y} \in \mathcal{C}(0, \overline{p}_{S_\tau})) - \Pr(\mathbf{Y} \in \mathcal{C}(0, \overline{p}_{S_{\tau-1}})) \tag{99}$$

$$= \Pr(\mathbf{Y} \in \mathcal{B}_{S_\tau}) - \Pr(\mathbf{Y} \in \mathcal{B}_{S_{\tau-1}}) \tag{100}$$

$$\leq \Pr(\mathbf{Y} \in \mathcal{B}_{S_\tau}) - \frac{(\tau - 1) \cdot \Pr(\mathbf{Y} \in \mathcal{B}_{S_\tau})}{\tau} \tag{101}$$

$$= \frac{\Pr(\mathbf{Y} \in \mathcal{B}_{S_\tau})}{\tau} \tag{102}$$

We obtain Equation 100 from Equation 99 based on Lemma 8, and Equation 101 from Equation 100 based on Equation 83. Therefore, we have the following:

$$\left(r_y(q_1', q_2) - \frac{\Pr(\mathbf{Y} \in \mathcal{B}_{S_\tau})}{\tau}\right) \cdot \left(r_y(q_1, q_2') - \frac{\Pr(\mathbf{Y} \in \mathcal{B}_{S_\tau})}{\tau}\right) \leq 0 \tag{103}$$

Thus, there exists $(q_1^\tau, q_2^\tau)$ such that $r_x(q_1^\tau, q_2^\tau) = \overline{p}_{b_\tau}, r_y(q_1^\tau, q_2^\tau) = \frac{\Pr(\mathbf{Y} \in \mathcal{B}_{S_\tau})}{\tau}$ based on Lemma 7. Then, we have the following based on the definition of $r_x, r_y$:

$$\Pr(\mathbf{X} \in \mathcal{C}'(q_1, q_2) \cap \mathcal{C}(q_1^\tau, q_2^\tau)) = \overline{p}_{b_\tau} \tag{104}$$

$$\Pr(\mathbf{Y} \in \mathcal{C}'(q_1, q_2) \cap \mathcal{C}(q_1^\tau, q_2^\tau)) = \frac{\Pr(\mathbf{Y} \in \mathcal{B}_{S_\tau})}{\tau} \tag{105}$$

Finally, we let $\mathcal{C}_{b_\tau} = \mathcal{C}'(q_1, q_2) \cap \mathcal{C}(q_1^\tau, q_2^\tau)$, which meets our goal.

**Step II:** Assuming we can find $\{(q_1^\tau, q_2^\tau), (q_1^{\tau-1}, q_2^{\tau-1}), \cdots, (q_1^{\tau-m+1}, q_2^{\tau-m+1})\}$ $(\forall j \in [\tau - m + 1, \tau], q_1 \le q_1^j \le q_2^j \le q_2)$ where $1 \le m \le \tau - 1$ such that $\forall j \in [\tau - m + 1, \tau]$, we have:

$$\Pr(\mathbf{X} \in \mathcal{C}_{b_j}) = \bar{p}_{b_j} \tag{106}$$

$$\Pr(\mathbf{Y} \in \mathcal{C}_{b_j}) = \frac{\Pr(\mathbf{Y} \in \mathcal{B}_{S_\tau})}{\tau}, \tag{107}$$

as well as the following:

$$\forall j, t \in [\tau - m + 1, \tau] \text{ and } j \ne t, \mathcal{C}_{b_j} \cap \mathcal{C}_{b_t} = \emptyset \tag{108}$$

We denote $e = \tau - m$. We show we can find $\mathcal{C}_{b_e}$ such that we have:

$$\Pr(\mathbf{X} \in \mathcal{C}_{b_e}) = \bar{p}_{b_e} \tag{109}$$

$$\Pr(\mathbf{Y} \in \mathcal{C}_{b_e}) = \frac{\Pr(\mathbf{Y} \in \mathcal{B}_{S_\tau})}{\tau} \tag{110}$$

$$\forall j \in [e + 1, \tau], \mathcal{C}_{b_e} \cap \mathcal{C}_{b_j} = \emptyset \tag{111}$$

We let $q_1 = 0, q_2 = \bar{p}_{S_\tau}$ and denote

$$\mathcal{C}'(q_1, q_2) = \mathcal{C}(q_1, q_2) \setminus \cup_{j=e+1}^\tau \mathcal{C}_{b_j} \tag{112}$$

We have the following:

$$r_x(q_1, q_2) \tag{113}$$

$$= \Pr(\mathbf{X} \in \mathcal{C}'(q_1, q_2) \cap \mathcal{C}(q_1, q_2)) \tag{114}$$

$$= \Pr(\mathbf{X} \in \mathcal{C}(q_1, q_2)) - \Pr(\mathbf{X} \in \cup_{j=e+1}^\tau \mathcal{C}_{b_j}) \tag{115}$$

$$= \Pr(\mathbf{X} \in \mathcal{C}(q_1, q_2)) - \sum_{j=e+1}^\tau \Pr(\mathbf{X} \in \mathcal{C}_{b_j}) \tag{116}$$

$$= \sum_{j=1}^\tau \bar{p}_{b_j} - \sum_{j=e+1}^\tau \bar{p}_{b_j} \tag{117}$$

$$= \sum_{j=1}^e \bar{p}_{b_j} \tag{118}$$

The Equation 116 from 115 is based on the Equation 108, and the Equation 117 from 116 is based the Equation 51 and 106. Furthermore, we have the following:

$$r_y(q_1, q_2) \tag{119}$$

$$= \Pr(\mathbf{Y} \in \mathcal{C}'(q_1, q_2) \cap \mathcal{C}(q_1, q_2)) \tag{120}$$

$$= \Pr(\mathbf{Y} \in \mathcal{C}(q_1, q_2)) - \Pr(\mathbf{Y} \in \cup_{j=e+1}^\tau \mathcal{C}_{b_j}) \tag{121}$$

$$= \Pr(\mathbf{Y} \in \mathcal{C}(q_1, q_2)) - \sum_{j=e+1}^\tau \Pr(\mathbf{Y} \in \mathcal{C}_{b_j}) \tag{122}$$

$$= \Pr(\mathbf{Y} \in \mathcal{B}_{S_\tau}) - \frac{m \cdot \Pr(\mathbf{Y} \in \mathcal{B}_{S_\tau})}{\tau} \tag{123}$$

$$= \frac{(\tau - m) \cdot \Pr(\mathbf{Y} \in \mathcal{B}_{S_\tau})}{\tau} \tag{124}$$

The Equation 123 from 122 is because $\mathcal{C}(q_1, q_2) = \mathcal{B}_{S_\tau}$ and the Equation 107. We have $\bar{p}_{b_e} \le r_x(q_1, q_2)$. Therefore, based on Lemma 7, there exist $q_1', q_2'$ such that:

$$r_x(q_1, q_2') = \bar{p}_{b_e} \tag{125}$$

$$r_x(q_1', q_2) = \bar{p}_{b_e} \tag{126}$$

We have:

$$r_y(q_1, q_2') \tag{127}$$

$$\geq r_y(q_1, q_2) \cdot \frac{1}{\lceil r_x(q_1, q_2)/r_x(q_1, q_2') \rceil} \tag{128}$$

$$\geq r_y(q_1, q_2) \cdot \frac{1}{\lceil (\sum_{j=1}^{e} \overline{p}_{b_j})/\overline{p}_{b_e} \rceil} \tag{129}$$

$$\geq \frac{\Pr(\mathbf{Y} \in \mathcal{B}_{S_\tau})}{\tau} \tag{130}$$

Equation 128 from 127 is based on Lemma 9, Equation 129 from 128 is based on Equation 90 and 125, and Equation 130 from 129 is obtained from Equation 91 and the fact that $\lceil (\sum_{j=1}^{e} \overline{p}_{b_j})/\overline{p}_{b_e} \rceil \geq \tau$. Next, we show:

$$r_y(q_1', q_2) \leq \frac{\Pr(\mathbf{Y} \in \mathcal{B}_{S_\tau})}{\tau} \tag{131}$$

In particular, we consider two scenarios.

*Scenario 1).* $q_1' \geq \min_{j=\tau-m+1}^{\tau} q_1^j$. We denote $t = \operatorname{argmin}_{j=\tau-m+1}^{\tau} q_1^j$. We let $\mathcal{C}_w = \mathcal{C}_{b_t} \subseteq \mathcal{C}(q_1^t, q_2^t)$. As $q_1' \geq q_1^t, q_2 \geq q_2^t$ and $r_x(q_1', q_2) = \overline{p}_{b_e} \leq \Pr(\mathbf{X} \in \mathcal{C}_w) = \overline{p}_{b_t}$, we have the following based on Lemma 10:

$$r_y(q_1', q_2) \leq \Pr(\mathbf{Y} \in \mathcal{C}_w) = \frac{\Pr(\mathbf{Y} \in \mathcal{B}_{S_\tau})}{\tau} \tag{132}$$

*Scenario 2).* $q_1' < \min_{j=\tau-m+1}^{\tau} q_1^j$. We have the following:

$$\mathcal{C}'(q_1, q_2) \cap \mathcal{C}(q_1, q_1') = \mathcal{C}(q_1, q_1') \tag{133}$$

Furthermore, we have:

$$r_x(q_1, q_1') \tag{134}$$

$$= r_x(q_1, q_2) - r_x(q_1', q_2) \tag{135}$$

$$= \sum_{j=1}^{e} \overline{p}_{b_j} - \overline{p}_{b_e} \tag{136}$$

$$= \sum_{j=1}^{e-1} \overline{p}_{b_j} \tag{137}$$

Moreover, we have $r_x(q_1, q_1') = q_1' - q_1$ from Lemma 8. The above two should be equal. Thus, we have $q_1' = \sum_{j=1}^{e-1} \overline{p}_{b_j} = \overline{p}_{S_{\tau-m-1}}$ since $e = \tau - m$. we have:

$$r_y(q_1', q_2) \tag{138}$$

$$= r_y(q_1, q_2) - r_y(q_1, q_1') \tag{139}$$

$$= r_y(q_1, q_2) - \Pr(\mathbf{Y} \in \mathcal{B}_{S_{\tau-m-1}}) \tag{140}$$

$$\leq \frac{(\tau - m) \cdot \Pr(\mathbf{Y} \in \mathcal{B}_{S_\tau})}{\tau} - \frac{(\tau - m - 1) \cdot \Pr(\mathbf{Y} \in \mathcal{B}_{S_\tau})}{\tau} \tag{141}$$

$$= \frac{\Pr(\mathbf{Y} \in \mathcal{B}_{S_\tau})}{\tau} \tag{142}$$

We obtain Equation 140 from Equation 139 based on Lemma 8.

Therefore, we have the following in both scenarios:

$$r_y(q_1, q_2') \geq \frac{\Pr(\mathbf{Y} \in \mathcal{B}_{S_\tau})}{\tau} \tag{143}$$

$$r_y(q_1', q_2) \leq \frac{\Pr(\mathbf{Y} \in \mathcal{B}_{S_\tau})}{\tau} \tag{144}$$

Based on Lemma 7, there exist $q_1^e, q_2^e$ such that $r_x(q_1^e, q_2^e) = \overline{p}_{b_\tau}, r_y(q_1^e, q_2^e) = \frac{\Pr(\mathbf{Y} \in \mathcal{B}_{S_\tau})}{\tau}$. Then, we have the following based on the definition of $r_x, r_y$:

$$\Pr(\mathbf{X} \in \mathcal{C}'(q_1, q_2) \cap \mathcal{C}(q_1^e, q_2^e)) = \overline{p}_{b_e} \tag{145}$$

$$\Pr(\mathbf{Y} \in \mathcal{C}'(q_1, q_2) \cap \mathcal{C}(q_1^e, q_2^e)) = \frac{\Pr(\mathbf{Y} \in \mathcal{B}_{S_\tau})}{\tau} \tag{146}$$

We let $\mathcal{C}_e = \mathcal{C}'(q_1, q_2) \cap \mathcal{C}(q_1^e, q_2^e)$. From the definition of $\mathcal{C}'(q_1, q_2)$, we have $\forall j \in [e + 1, \tau], \mathcal{C}'(q_1, q_2) \cap \mathcal{C}_{b_j} = \emptyset$. Thus, we have $\forall j \in [e + 1, \tau], \mathcal{C}_{b_e} \cap \mathcal{C}_{b_j} = \emptyset$ since $\mathcal{C}_{b_e} \subseteq \mathcal{C}'(q_1, q_2)$.

Therefore, we reach our goal by Mathematical Induction, i.e., for $\forall j \in [1, \tau]$, we have:

$$\Pr(\mathbf{X} \in \mathcal{C}_{b_j}) = \overline{p}_{b_j} \tag{147}$$

$$\Pr(\mathbf{Y} \in \mathcal{C}_{b_j}) = \frac{\Pr(\mathbf{Y} \in \mathcal{B}_{S_\tau})}{\tau} \tag{148}$$

We can also verify that $\cup_{j=1}^\tau \mathcal{C}_{b_j} = \mathcal{B}_{S_\tau}$.

Next, we show our proof based on Mathematical Induction for the other part, i.e., $\mathcal{B}_{S_k} \setminus \mathcal{B}_{S_\tau}$. Our construction process is similar to the above first part but has subtle differences.

**Step I:** Let $q_1 = \sum_{j=1}^\tau \overline{p}_{b_j}$ and $q_2 = \sum_{j=1}^k \overline{p}_{b_j}$. We define:

$$\mathcal{C}'(q_1, q_2) = \mathcal{C}(q_1, q_2) = \mathcal{B}_{S_k} \setminus \mathcal{B}_{S_\tau} \tag{149}$$

Then, we have:

$$r_x(q_1, q_2) = q_2 - q_1 = \sum_{j=\tau+1}^k \overline{p}_{b_j} \tag{150}$$

$$r_y(q_1, q_2) \tag{151}$$
$$= \Pr(\mathbf{Y} \in \mathcal{C}'(q_1, q_2) \cap \mathcal{C}(q_1, q_2)) \tag{152}$$
$$= \Pr(\mathbf{Y} \in \mathcal{C}(0, q_2)) - \Pr(\mathbf{Y} \in \mathcal{C}(q_1, q_2)) \tag{153}$$
$$= \Pr(\mathbf{Y} \in \mathcal{B}_{S_k}) - \Pr(\mathbf{Y} \in \mathcal{B}_{S_\tau}) \tag{154}$$
$$\geq \frac{k \cdot \Pr(\mathbf{Y} \in \mathcal{B}_{S_\tau})}{\tau} - \Pr(\mathbf{Y} \in \mathcal{B}_{S_\tau}) \tag{155}$$
$$= \frac{(k - \tau) \cdot \Pr(\mathbf{Y} \in \mathcal{B}_{S_\tau})}{\tau} \tag{156}$$

The Equation 150 is based on the fact that $\mathcal{C}'(q_1, q_2) = \mathcal{C}(q_1, q_2)$ and Definition 1, and we obtain Equation 154 from 155 based on Equation 83. We have $\overline{p}_{b_k} \leq r_x(q_1, q_2)$. Therefore, based on Lemma 7, we know that there exists $q_1' = q_2 - \overline{p}_{b_k}, q_2' = q_1 + \overline{p}_{b_k}$ such that:

$$r_x(q_1, q_2') = \overline{p}_{b_k} \tag{157}$$
$$r_x(q_1', q_2) = \overline{p}_{b_k} \tag{158}$$

We consider two scenarios.

*Scenario 1).* In this scenario, we consider $r_y(q_1', q_2) > \frac{\Pr(\mathbf{Y} \in \mathcal{B}_{S_\tau})}{\tau}$. We let $q_1^k = q_1', q_2^k = q_2$, i.e., we have $\mathcal{C}_{b_k} = \mathcal{C}(q_1', q_2) \cap \mathcal{C}'(q_1, q_2)$. Then, we have:

$$\Pr(\mathbf{X} \in \mathcal{C}_{b_k}) = r_x(q_1', q_2) = \overline{p}_{b_k} \tag{159}$$

$$\Pr(\mathbf{Y} \in \mathcal{C}_{b_k}) = r_y(q_1', q_2) \geq \frac{\Pr(\mathbf{Y} \in \mathcal{B}_{S_\tau})}{\tau} \tag{160}$$

Therefore, we have the following:

$$\Pr(\mathbf{Y} \in \mathcal{C}(q_1, q_2) \setminus \mathcal{C}_{b_k}) \tag{161}$$
$$= \Pr(\mathbf{Y} \in \mathcal{C}(q_1, q_1')) \tag{162}$$
$$= \Pr(\mathbf{Y} \in \mathcal{C}(0, q_1')) - \Pr(\mathbf{Y} \in \mathcal{C}(0, q_1)) \tag{163}$$
$$= \Pr(\mathbf{Y} \in \mathcal{B}_{S_{k-1}}) - \Pr(\mathbf{Y} \in \mathcal{B}_{S_\tau}) \tag{164}$$
$$\geq \frac{(k - \tau - 1) \cdot \Pr(\mathbf{Y} \in \mathcal{B}_{S_\tau})}{\tau} \tag{165}$$

*Scenario 2).* In this scenario, we consider $r_y(q'_1, q_2) \leq \frac{\Pr(\mathbf{Y} \in \mathcal{B}_{S_\tau})}{\tau}$. We have the following:

$$r_y(q_1, q'_2) \tag{166}$$

$$\geq r_y(q_1, q_2) \cdot \frac{1}{\lceil r_x(q_1, q_2)/r_x(q_1, q'_2) \rceil} \tag{167}$$

$$\geq \frac{(k - \tau) \cdot \Pr(\mathbf{Y} \in \mathcal{B}_{S_\tau})}{\tau} \cdot \frac{1}{k - \tau} \tag{168}$$

$$= \frac{\Pr(\mathbf{Y} \in \mathcal{B}_{S_\tau})}{\tau} \tag{169}$$

We obtain Equation 167 from 166 via Lemma 9, and we obtain Equation 168 from 167 based on Equation 150 to 156 and the fact $r_x(q_1, q_2)/r_x(q_1, q'_2) = \frac{\sum_{j=\tau+1}^{k} \overline{p}_{b_j}}{\overline{p}_{b_{\tau+1}}} \geq k - \tau$. We have $(r_y(q'_1, q_2) - \frac{\Pr(\mathbf{Y} \in \mathcal{B}_{S_\tau})}{\tau}) \cdot (r_y(q_1, q'_2) - \frac{\Pr(\mathbf{Y} \in \mathcal{B}_{S_\tau})}{\tau}) \leq 0$. Therefore, from Lemma 7, we know that there exist $(q_1^k, q_2^k)$ such that:

$$\Pr(\mathbf{X} \in \mathcal{C}_{b_k}) = r_x(q_1^k, q_2^k) = \overline{p}_{b_k} \tag{170}$$

$$\Pr(\mathbf{Y} \in \mathcal{C}_{b_k}) = r_y(q_1^k, q_2^k) = \frac{\Pr(\mathbf{Y} \in \mathcal{B}_{S_\tau})}{\tau} \tag{171}$$

We also have the following:

$$\Pr(\mathbf{Y} \in \mathcal{C}(q_1, q_2) \setminus \mathcal{C}_{b_k}) \tag{172}$$

$$= r_y(q_1, q_2) - r_y(q_1^k, q_2^k) \tag{173}$$

$$\geq \frac{(k - \tau - 1) \cdot \Pr(\mathbf{Y} \in \mathcal{B}_{S_\tau})}{\tau} \tag{174}$$

Similarly, we let $\mathcal{C}_{b_k} = \mathcal{C}'(q_1, q_2) \cap \mathcal{C}(q_1^k, q_2^k)$.

Based on the conditions of our constructions in the two scenarios, we know that if $\Pr(\mathbf{Y} \in \mathcal{B}_{b_k}) > \frac{\Pr(\mathbf{Y} \in \mathcal{B}_{S_\tau})}{\tau}$, then we have $q_2^k = q_2$.

**Step II:** We show that if we can find $(q_1^k, q_2^k), \cdots, (q_1^{k-m+1}, q_2^{k-m+1})$ where $m \in [1, k - \tau - 1]$ and $\mathcal{C}_{b_j}, \forall j \in [k, k - m + 1]$ such that:

$$\Pr(\mathbf{X} \in \mathcal{C}_{b_j}) = \overline{p}_{b_j} \tag{175}$$

$$\Pr(\mathbf{Y} \in \mathcal{C}_{b_j}) \geq \frac{\Pr(\mathbf{Y} \in \mathcal{B}_{S_\tau})}{\tau} \tag{176}$$

$$\Pr(\mathbf{Y} \in \mathcal{C}(q_1, q_2) \setminus \cup_{t=k-m+1}^{k} \mathcal{C}_{b_t}) \geq \frac{(k - \tau - m) \cdot \Pr(\mathbf{Y} \in \mathcal{B}_{S_\tau})}{\tau} \tag{177}$$

Then, we can find $(q_1^{k-m}, q_2^{k-m})$ such that:

$$\Pr(\mathbf{X} \in \mathcal{C}_{b_{k-m}}) = \overline{p}_{b_{k-m}} \tag{178}$$

$$\Pr(\mathbf{Y} \in \mathcal{C}_{b_{k-m}}) \geq \frac{\Pr(\mathbf{Y} \in \mathcal{B}_{S_\tau})}{\tau} \tag{179}$$

$$\Pr(\mathbf{Y} \in \mathcal{C}(q_1, q_2) \setminus \cup_{t=k-m}^{k} \mathcal{C}_{b_t}) \geq \frac{(k - \tau - m - 1) \cdot \Pr(\mathbf{Y} \in \mathcal{B}_{S_\tau})}{\tau} \tag{180}$$

For simplicity, we denote $e = k - m$, we let $q_1 = \sum_{j=1}^{\tau} \overline{p}_{b_j}$ and $q_2 = \sum_{j=1}^{k} \overline{p}_{b_j}$, and we define:

$$\mathcal{C}'(q_1, q_2) = \mathcal{C}(q_1, q_2) \setminus \cup_{j=e+1}^{k} \mathcal{C}_{b_j} \tag{181}$$

Then, we have:

$$r_x(q_1, q_2) = \Pr(\mathbf{X} \in \mathcal{C}(q_1, q_2) \setminus \cup_{j=e+1}^{k} \mathcal{C}_{b_j}) = \sum_{j=\tau+1}^{e} \overline{p}_{b_j} \tag{182}$$

$$r_y(q_1, q_2) = \Pr(\mathbf{Y} \in \mathcal{C}(q_1, q_2) \setminus \cup_{j=e+1}^{k} \mathcal{C}_{b_j}) \geq \frac{(k - \tau - m) \cdot \Pr(\mathbf{Y} \in \mathcal{B}_{S_\tau})}{\tau} \tag{183}$$

We have $\overline{p}_{b_e} \le r_x(q_1, q_2)$. Therefore, based on Lemma 7, we know that there exists $q_1', q_2'$ such that:

$$r_x(q_1', q_2) = \overline{p}_{b_e} \tag{184}$$

$$r_x(q_1, q_2') = \overline{p}_{b_e} \tag{185}$$

Similarly, we consider two scenarios:

*Scenario 1).* In this scenario, we consider that the following holds:

$$r_y(q_1', q_2) > \frac{\Pr(\mathbf{Y} \in \mathcal{B}_{S_\tau})}{\tau} \tag{186}$$

We let $q_1^e = q_1', q_2^e = q_2$, i.e., $\mathcal{C}_{b_e} = \mathcal{C}(q_1', q_2) \cap \mathcal{C}'(q_1, q_2)$. Then, we have:

$$\Pr(\mathbf{X} \in \mathcal{C}_{b_e}) = r_x(q_1', q_2) = \overline{p}_{b_e} \tag{187}$$

$$\Pr(\mathbf{Y} \in \mathcal{C}_{b_e}) = r_y(q_1', q_2) \ge \frac{\Pr(\mathbf{Y} \in \mathcal{B}_{S_\tau})}{\tau} \tag{188}$$

We note that we have $q_1' \le \min_{j=e+1}^{k} q_1^i$ in this scenario. Otherwise, Equation 186 will not hold based on Lemma 10. We give a short proof.

Assuming $q_1' > \min_{j=k-m+1}^{k} q_1^j$. We denote $w = \operatorname{argmin}_{j=k-m+1}^{k} q_1^j$. We let $\mathcal{C}_w = \mathcal{C}_{b_w} \subseteq \mathcal{C}(q_1^w, q_2^w)$. Note that in this case, we have $q_2^w < q_2$ because $q_2^w = q_2$ and $q_1' > q_1^w$ cannot hold at the same time as long as $r_y(q_1', q_2) > 0$. Thus, we have $\Pr(\mathbf{Y} \in \mathcal{C}_w) = \frac{\Pr(\mathbf{Y} \in \mathcal{B}_{S_\tau})}{\tau}$ because if $\Pr(\mathbf{Y} \in \mathcal{C}_w) > \frac{\Pr(\mathbf{Y} \in \mathcal{B}_{S_\tau})}{\tau}$, we have $q_2^w = q_2$. As we have $q_1' > q_1^w, q_2 > q_2^w$ and $r_x(q_1', q_2) = \overline{p}_{b_e} \le \Pr(\mathbf{X} \in \mathcal{C}_w) = \overline{p}_{b_w}$. We have the following based on Lemma 10:

$$r_y(q_1', q_2) \le \Pr(\mathbf{Y} \in \mathcal{C}_w) = \frac{\Pr(\mathbf{Y} \in \mathcal{B}_{S_\tau})}{\tau} \tag{189}$$

Since Equation 186 and Equation 189 cannot hold at the same time, the assumption $q_1' > \min_{j=k-m+1}^{k} q_1^j$ must be wrong. Therefore, we have $q_1' \le \min_{j=k-m+1}^{k} q_1^j$.

Based on $q_1' \le \min_{j=k-m+1}^{k} q_1^j$, we have the following:

$$\mathcal{C}'(q_1, q_2) \cap \mathcal{C}(q_1, q_1') = \mathcal{C}(q_1, q_1') \tag{190}$$

Therefore, we have $r_x(q_1, q_1') = q_1' - q_1$ from Definition 1. Moreover, we have the following:

$$r_x(q_1, q_1') \tag{191}$$

$$= r_x(q_1, q_2) - r_x(q_1', q_2) \tag{192}$$

$$= \sum_{j=\tau+1}^{e} \overline{p}_{b_j} - \overline{p_{b_e}} \tag{193}$$

$$= \sum_{j=\tau+1}^{e-1} \overline{p}_{b_j} \tag{194}$$

The above two should be equal. Therefore, we have $q_1' = \sum_{j=\tau+1}^{e-1} p_{b_j} + q_1 = \sum_{j=1}^{e-1} p_{b_j}$. Recall that we let $\mathcal{C}_{b_e} = \mathcal{C}'(q_1, q_2) \cap \mathcal{C}(q_1', q_2)$. Thus, we have:

$$\Pr(\mathbf{Y} \in \mathcal{C}(q_1, q_2) \setminus \cup_{j=e}^{k} \mathcal{C}_{b_j}) \tag{195}$$

$$= \Pr(\mathbf{Y} \in \mathcal{C}'(q_1, q_2) \setminus \mathcal{C}_{b_e}) \tag{196}$$

$$= \Pr(\mathbf{Y} \in \mathcal{C}(q_1, q_1')) \tag{197}$$

$$= \Pr(\mathbf{Y} \in \mathcal{C}(0, q_1')) - \Pr(\mathbf{Y} \in \mathcal{C}(0, q_1)) \tag{198}$$

$$= \Pr(\mathbf{Y} \in \mathcal{B}_{S_{e-1}}) - \Pr(\mathbf{Y} \in \mathcal{B}_{S_\tau}) \tag{199}$$

$$\ge \frac{(k - \tau - m - 1) \cdot \Pr(\mathbf{Y} \in \mathcal{B}_{S_\tau})}{\tau} \tag{200}$$

*Scenario 2).* In this scenario, we consider that the following holds:

$$r_y(q_1', q_2) \le \frac{\Pr(\mathbf{Y} \in \mathcal{B}_{S_\tau})}{\tau} \tag{201}$$

Note that we have:

$$r_y(q_1, q_2') \tag{202}$$

$$\geq r_y(q_1, q_2) \cdot \frac{1}{\lceil r_x(q_1, q_2)/r_x(q_1, q_2') \rceil} \tag{203}$$

$$\geq r_y(q_1, q_2) \cdot \frac{1}{k - \tau - m} \tag{204}$$

$$\geq \frac{\Pr(\mathbf{Y} \in \mathcal{B}_{S_\tau})}{\tau} \tag{205}$$

We obtain Equation 203 from 202 via Lemma 9, and we obtain Equation 204 from 203 based on Equation 183 and the fact $r_x(q_1, q_2)/r_x(q_1, q_2') = \frac{\sum_{j=\tau+1}^{e} \overline{p}_{b_j}}{\overline{p}_{b_{\tau+1}}} \geq k - \tau - m$. We have $(r_y(q_1', q_2) - \frac{\Pr(\mathbf{Y} \in \mathcal{B}_{S_\tau})}{\tau}) \cdot (r_y(q_1, q_2') - \frac{\Pr(\mathbf{Y} \in \mathcal{B}_{S_\tau})}{\tau}) \leq 0$. Based on Lemma 7, we can find $(q_1^e, q_2^e)$ such that we have:

$$\Pr(\mathbf{X} \in \mathcal{C}'(q_1, q_2) \cap \mathcal{C}(q_1^e, q_2^e)) = r_x(q_1^e, q_2^e) = \overline{p}_{b_e} \tag{206}$$

$$\Pr(\mathbf{Y} \in \mathcal{C}'(q_1, q_2) \cap \mathcal{C}(q_1^e, q_2^e)) = r_y(q_1^e, q_2^e) = \frac{\Pr(\mathbf{Y} \in \mathcal{B}_{S_\tau})}{\tau} \tag{207}$$

We let $\mathcal{C}_{b_e} = \mathcal{C}'(q_1, q_2) \cap \mathcal{C}(q_1^e, q_2^e)$. We also have the following:

$$\Pr(\mathbf{Y} \in \mathcal{C}(q_1, q_2) \setminus \cup_{j=e}^{k} \mathcal{C}_{b_j}) \tag{208}$$

$$= \Pr(\mathbf{Y} \in \mathcal{C}'(q_1, q_2) \setminus \mathcal{C}_{b_e}) \tag{209}$$

$$= r_y(q_1, q_2) - r_y(q_1^e, q_2^e) \tag{210}$$

$$\geq \frac{(k - \tau - m) \cdot \Pr(\mathbf{Y} \in \mathcal{B}_{S_\tau})}{\tau} - \frac{\Pr(\mathbf{Y} \in \mathcal{B}_{S_\tau})}{\tau} \tag{211}$$

$$\geq \frac{(k - \tau - m - 1) \cdot \Pr(\mathbf{Y} \in \mathcal{B}_{S_\tau})}{\tau} \tag{212}$$

Similar to **Step I**, we still hold the conclusion that if $\Pr(\mathbf{Y} \in \mathcal{C}_{b_e}) > \frac{\Pr(\mathbf{Y} \in \mathcal{B}_{S_\tau})}{\tau}$, we have $q_2^e = q_2$. Then, we can apply *Mathematical Induction* to reach the conclusion. Also, we can verify $\cup_{j=\tau+1}^{k} \mathcal{C}_{b_j} = \mathcal{B}_{S_k} \setminus \mathcal{B}_{S_\tau}$. □

## C  PROOF OF PROPOSITION 1

The function SAMPLEUNDERNOISE$(f, k, \sigma, \mathbf{x}, n, \alpha)$ works as follows: we first draw $n$ random noise from $\mathcal{N}(0, \sigma^2 I)$, i.e., $\epsilon_1, \epsilon_2, \cdots, \epsilon_n$. Then, we compute the values: $\forall i \in [1, c], c_i = \sum_{j=1}^{n} \mathbb{I}(f(\mathbf{x} + \epsilon_j) = i)$. The function BINOMPVALUE$(n_{c_t}, n_{c_t} + n_{c_{t+1}}, p)$ returns the result of p-value of the two-sided hypothesis test for $n_{c_t} \sim Bin(n_{c_t} + n_{c_{t+1}}, p)$.

**Proposition 1:** With probability at least $1 - \alpha$ over the randomness in PREDICT, if PREDICT returns a set $T$ (i.e., does not ABSTAIN), then we have $g_k(\mathbf{x}) = T$.

*Proof.* We aim to compute the probability that PREDICT returns a set which not equals to $g_k(\mathbf{x})$, which happens if and only if $g_k(\mathbf{x}) \neq T$ and PREDICT doesn't abstain. Specifically, we have:

$$\Pr(\text{PREDICT returns a set } \neq g_k(\mathbf{x})) \tag{213}$$

$$= \Pr(g_k(\mathbf{x}) \neq T, \text{PREDICT doesn't abstain}) \tag{214}$$

$$= \Pr(g_k(\mathbf{x}) \neq T) \cdot \Pr(\text{PREDICT doesn't abstain}|g_k(\mathbf{x}) \neq T) \tag{215}$$

$$\leq \Pr(\text{PREDICT doesn't abstain}|g_k(\mathbf{x}) \neq T) \tag{216}$$

Theorem 1 in Hung et al. (2019) shows the above conditional probability is as follows:

$$\Pr(\text{PREDICT doesn't abstain}|g_k(\mathbf{x}) \neq T) \leq \alpha \tag{217}$$

Therefore, we reach the conclusion. □

## D    PROOF OF PROPOSITION 2

**Proposition 2:**  With probability at least $1-\alpha$ over the randomness in CERTIFY, if CERTIFY returns a radius $\underline{R_l}$ (i.e., does not ABSTAIN), then we have $l \in g_k(\mathbf{x}+\delta), \forall \|\delta\|_2 < \underline{R_l}$.

*Proof.* From the definition of BINOCP and SIMUEM, we know the probability that the following inequalities simultaneously hold is at least $1-\alpha$ over the sampling of counts:

$$\underline{p_i} \leq \Pr(f(x+\epsilon)=i) \text{ if } i = l \tag{218}$$

$$\overline{p_i} \geq \Pr(f(x+\epsilon)=i) \text{ if } i \neq l \tag{219}$$

Then, with the returned bounds, we can invoke Theorem 1 to obtain the robustness guarantee if the calculated radius is larger than 0. Note that otherwise CERTIFY abstains.  □

## E    CERTIFIED TOP-$k$ ACCURACY

We show how to derive a lower bound of the certified top-$k$ accuracy based on the approximate certified top-$k$ accuracy. The process is similar to that Cohen et al. (2019) used to derive a lower bound of the certified top-1 accuracy based on the approximate certified top-1 accuracy. Specifically, we have the following lemma from Cohen et al. (2019).

**Lemma 11.**  *Let $z_i$ be a binary variable and $Y_i$ be a Bernoulli random variable. Suppose if $z_i = 1$, then $Pr(Y_i = 1) \leq \alpha$. Then, for any $\rho > 0$, with probability at least $1 - \rho$, we have the following:*

$$\frac{1}{m}\sum_{i=1}^{m} z_i \geq \frac{1}{1-\alpha}\left(\frac{\sum_{i=1}^{m} Y_i}{m} - \alpha - \sqrt{\frac{2\alpha(1-\alpha)\log\frac{1}{\rho}}{m}} - \frac{\log(\frac{1}{\rho})}{3m}\right) \tag{220}$$

*Proof.*  Please refer to Cohen et al. (2019).  □

Assuming we have a test dataset $D_{test} = \{(\mathbf{x}_1, y_1), (\mathbf{x}_2, y_2), \cdots, (\mathbf{x_m}, y_m)\}$ as well as a radius $r$. We define the following indicate value:

$$a_i = \mathbb{I}(y_i \in g_k(\mathbf{x}_i+\delta)), \forall\|\delta\|_2 < r \tag{221}$$

Then, the certified top-$k$ accuracy of the smoothed classifier $g$ at radius $r$ can be computed as $\frac{1}{m}\sum_{i=1}^{m} a_i$. For each sample $\mathbf{x}_i$, we run the CERTIFY function with $1-\alpha$ confidence level and we use a random variable $Y_i$ to denote that the function CERTIFY returns a radius bigger than $r$. From **Proposition 2**, we know:

$$\Pr(Y_i = 1) \leq \alpha, \text{ if } a_i = 1 \tag{222}$$

The approximate certified top-$k$ accuracy of the smoothed classifier at radius $r$ is $\frac{1}{m}\sum_{i=1}^{m} Y_i$. Then, we can use Lemma 11 to obtain a lower bound of $\frac{1}{m}\sum_{i=1}^{m} a_i$. Specifically, for any $\rho > 0$, with probability at least $1-\rho$ over the randomness of CERTIFY, we have:

$$\frac{1}{m}\sum_{i=1}^{m} a_i \geq \frac{1}{1-\alpha}\left(\frac{\sum_{i=1}^{m} Y_i}{m} - \alpha - \sqrt{\frac{2\alpha(1-\alpha)\log\frac{1}{\rho}}{m}} - \frac{\log(\frac{1}{\rho})}{3m}\right) \tag{223}$$

We can see that the difference between the certified top-$k$ accuracy and the approximate certified top-$k$ accuracy is negligible when $\alpha$ is small.

