# OpenReview forum: "Certified Robustness for Top-k Predictions against Adversarial Perturbations via Randomized Smoothing"
_ICLR.cc/2020/Conference — Accept (Poster)_

### Official Review · AnonReviewer3 · 2019-10-22
**Official Blind Review #3**

**Rating:** 6

**Review:**

Summary.
The paper proposes an extension to the work of Cohen et al. where a certified radius is deduced using a randomized smoothing approach. In particular, the authors show the radius at which a smoothed classifier g at under Gaussian perturbations is certified for the top k predictions. That is to say that the prediction will remain within the top k predictions of g. Setting k=1, one recovers Cohen et al. results. The authors show that the derived radius is tight.

Comments.

I generally find the work interesting and I do not have any major criticism. The paper is also easy to read. I did not go through the tightness proof but I skimmed through proof of the certified radius and I find that the argument follows in a similar fashion to previous works using NP lemma.

May the authors comment on the following.

a) How does the method scale with k? - speedwise particularly when estimating the lower and upper bounds of the output probabilities.

b) I do not understand Figure 3. Can the authors comment why for the radius = 0 the certified accuracy of the larger sigma (1.0) is actually worse than the smaller sigma? At least when k=1, increasing sigma increases the certified radius in which I expect to see that most of the samples to be actually within the radius and it should perform much better than lower sigma {0.25,0.5}.

c) Based on the previous comment, I find the performance in Figure 3 in not consistent with the authors' discussion in page 7 "when sigma is larger, the accuracy under no attacks (i.e. the accuracy when radius is 9) is larger .. ".

d) Authors use bisection to find the solution to the certified accuracy for every t. Bisection is known to only enjoy linear convergence rate. Have the authors considered using algorithms that are much faster such as the sueprlinear secant method?

e) Page 3 bullet point 3. "it is impossible to certify a L_2 radius" >> "it is impossible to certify an \ell_2 radius". Consider changing all L_2 to \ell_2.

**Experience Assessment:**

I have published one or two papers in this area.

**Review Assessment: Checking Correctness Of Derivations And Theory:**

I assessed the sensibility of the derivations and theory.

**Review Assessment: Checking Correctness Of Experiments:**

I carefully checked the experiments.

**Review Assessment: Thoroughness In Paper Reading:**

I read the paper thoroughly.

---

> ### Author Response · Authors · 2019-11-06
> **Author response**
>
> Thanks for the reviews. Our responses to the comments are as follows:
>
> Response 1: Given k, we can estimate the lower and upper bounds of the output probabilities only for the top-k labels with the largest frequencies among the random samples. Therefore, a larger k leads to estimation of probability bounds for more labels. However, such computational overhead is negligible. For instance, even if we compute the probability bounds for all labels, the time spent on estimating the probability bounds is less than 1% of the overall computation time.
>
> Response 2: radius=0 corresponds to the accuracy under no attack, which decreases as \sigma increases. The reason is that we add larger noise during training and testing when \sigma is larger, which decreases the accuracy under no attack.
>
> Response 3: Sorry for the confusion. It was a typo in our description. The description should be “when \sigma is smaller...”. We modified the sentence in the paper to be “Specifically, when \sigma  is smaller, the accuracy under no attacks (i.e., the accuracy when radius is 0) is larger, but the certified top-k accuracy drops more quickly as the radius increases.”.
>
> Response 4: Thanks for the suggestion! However, we found that the time for finding the solution to the certified accuracy is negligible, compared to the overall computation time of randomized smoothing. Specifically, the major computation time is for the base classifier to predict labels for the randomly perturbed inputs.
>
> Response 5: We have updated all L_2 to \ell_2 in the paper.

---

### Official Review · AnonReviewer1 · 2019-10-23
**Official Blind Review #1**

**Rating:** 6

**Review:**

This paper builds upon the random smoothing technique for top-1 prediction proposed by Cohen et al. for certifying top-k predictions with probabilistic guarantees, which enjoys good scalability to large neural networks and in principle can be applied to any classifier.

- Contributions:
1. The authors aim to provide (probabilistic) certification on top-k predictions, which to my knowledge is the first work to consider this setup. Many applications such as recommendation systems indeed use top-k predictions as a performance measure. The problem setup is new and important in the research of robustness certification.

2. In terms of technical contributions, the authors identify the difficulty of extending top-1 prediction to top-k prediction, due to the requirement of simultaneous confidence interval estimation of the bounds on the actual class predictions. To cope with this difficulty, the authors proposed simultaneous confidence interval estimation based on Clopper-Pearson method and Bonferroni correction. However, I am not sure the difficulty is caused by the necessity of estimating multiple probability bounds, or simply the limitation of the proposed algorithm. I hope the authors can address my concerns in the Questions below.

3. Experimental results on Cifar-10 and ImageNet showed improved lower bound on certified L2-norm radius when increasing k. The authors also performed an ablation study of different parameters in the proposed algorithm.

- Questions:
1. Intuitively, when extending top-1 certification to top-k certification, one would expect using ordered statistics of the prediction outputs from the randomly perturbed inputs. As long as the original label's prediction probability is in the top-k label set, the smoothed classifier is directly certified. Instead of ordered statistics, the authors tackle this problem by considering estimating upper and lower bounds of each class prediction probability. Therefore, the problem becomes more difficult as k increases, since this indirect approach needs to simultaneous estimate those probability bounds. I wonder the current approach will be suboptimal when compared to the ordered statistics approach. I would like to know the authors thoughts on this regard. That is, is the claimed difficulty an outcome when using the proposed indirect bound estimation for certification, or it's provably more difficult?

2. The discussion on Fig.3 says "We observe that  \sigma controls a trade-off between normal accuracy under no attacks and robustness. Specifically, when  is larger, the accuracy under no attacks (i.e., the accuracy when radius is 0) is larger, but the certified top-k accuracy drops more quickly as the radius increases." However, it seems that larger \sigma actually gives lower accuracy under no attacks in Figure 3. Please clarify.

Overall, this paper brings some new insights and results in robustness certification, but some claims and statements need to be further justified. I am happy to increase my rating if my concerns are addressed.


**Experience Assessment:**

I have published in this field for several years.

**Review Assessment: Checking Correctness Of Derivations And Theory:**

I assessed the sensibility of the derivations and theory.

**Review Assessment: Checking Correctness Of Experiments:**

I assessed the sensibility of the experiments.

**Review Assessment: Thoroughness In Paper Reading:**

I read the paper thoroughly.

---

> ### Author Response · Authors · 2019-11-06
> **Author response**
>
> Thanks for the reviews. Our responses to the comments are as follows:
>
> Response 1: The challenge for ordered statistics is how to derive the certified radius. Given an example and some randomly sampled noise, ordered statistics can probabilistically estimate whether a label is among the top-k labels predicted by the smoothed classifier. Ordered statistics can be used to estimate the top-k labels predicted by the smoothed classifier for an example with probabilistic guarantees. However, it is challenging for ordered statistics to derive the certified radius. Specifically, it is challenging to determine the upper bound of the adversarial perturbation (i.e., certified radius), with which a label is still among the top-k predicted labels. We believe it is an interesting future work to study how to leverage ordered statistics to derive certified radius.
>
> Response 2: Sorry for the confusion. It was a typo in our description. The description should be “when \sigma is smaller...”. We modified the sentence to be “Specifically, when \sigma  is smaller, the accuracy under no attacks (i.e., the accuracy when radius is 0) is larger, but the certified top-k accuracy drops more quickly as the radius increases.”

---

> > ### Comment · AnonReviewer1 · 2019-11-11
> > **Thank the authors for the response**
> >
> > I thank the authors for the response, which clearly addresses my concerns. I will increase my rating.

---

> > > ### Author Response · Authors · 2019-11-12
> > > **Thank the reviewer for the comments**
> > >
> > > We thank the reviewer for the insightful comments and carefully reading of our responses.

---

### Official Review · AnonReviewer2 · 2019-10-23
**Official Blind Review #2**

**Rating:** 6

**Review:**

Summary:

This paper studies the certifiable bounds for adversarial perturbations in \ell_2 radius for top-k predictions instead of top-1 predictions.  The paper obtains a certifiable radius of \ell_2 perturbations in the case of top-k predictions (Theorem 1) and shows that the bounds are tight (Theorem 2). The result thus generalizes the results obtained in Cohen et al. (2019) by setting k=1.
Since Theorem 1 requires lower and upper bounds, the paper proposes two methods for calculating the bounds on multinomial probabilities. Experimental evidence suggests that one indeed obtains a better certifiable radius for the top-k radius vs. the top-1 radius.

My evaluation of the paper is positive: The theoretical results (Theorem 1 and 2) are new and study practical use cases of these models. The experimental results (Figure 1) support the claim that there is a non-trivial difference between the certified radii of top-1 and top-k predictions.
However, the level of technical novelty is relatively low. The proof of Theorem 1 follows the similar procedure for top-1 predictions and the methods proposed for estimating probabilities (BinoCP and SinuEM) are standard procedures.

Other comments:

1. What is the trend of top-k-clean-accuracy and top-k-adversarial-accuracy as a function of k? Is this trend similar across different radii?

2. What is the value of k in Figure 3 and Figure 4?

**Experience Assessment:**

I have read many papers in this area.

**Review Assessment: Checking Correctness Of Derivations And Theory:**

I assessed the sensibility of the derivations and theory.

**Review Assessment: Checking Correctness Of Experiments:**

I assessed the sensibility of the experiments.

**Review Assessment: Thoroughness In Paper Reading:**

I read the paper at least twice and used my best judgement in assessing the paper.

---

> ### Author Response · Authors · 2019-11-06
> **Author response**
>
> Thanks for the reviews. Our responses to the comments are as follows:
>
> Response 1: On CIFAR10, the gaps between the certified top-k accuracy for different k are smaller than those between the top-k accuracy under no attacks (i.e., top-k-clean-accuracy), and they become smaller as the radius increases. On ImageNet, the gaps between the certified top-k accuracy for different k remain similar to those between the top-k accuracy under no attacks as the radius increases. Please refer to Figure 1. We added the analysis to Section 4.2.
>
> Response 2: k is 3 in these two figures. We modified the captions of Figure 3 and Figure 4 to indicate the value of k.

---

### Decision · Program_Chairs · 2019-12-19

**Decision:**

Accept (Poster)

**Comment:**

The paper extends the work on randomized smoothing for certifiably robust classifiers developed in prior work to a weaker specification requiring that the set of top-k predictions remain unchanged under adversarial perturbations of the input (rather than just the top-1). This enables the authors to achieve stronger results on robustness of classifiers on CIFAR10 and ImageNet (where the authors report the top-5 accuracy).

This is an interesting extension of certified defenses that is likely to be relevant for complex prediction tasks with several classes (ImageNet and beyond), where top-1 robustness may be difficult and unrealistic to achieve.

The reviewers were in consensus on acceptance and minor concerns were alleviated during the rebuttal phase.

I therefore recommend acceptance.